# Validation of human telomere length multi-ancestry meta-analysis association signals identifies *POP5* and *KBTBD6* as human telomere length regulation genes

Genome-wide association studies (GWAS) have become well-powered to detect loci associated with telomere length. However, no prior work has validated genes nominated by GWAS to examine their role in telomere length regulation. We conducted a multi-ancestry meta-analysis of 211,369 individuals and identified five novel association signals. Enrichment analyses of chromatin state and cell-type heritability suggested that blood/immune cells are the most relevant cell type to examine telomere length association signals. We validated specific GWAS associations by overexpressing *KBTBD6* or *POP5* and demonstrated that both lengthened telomeres. CRISPR/Cas9 deletion of the predicted causal regions in K562 blood cells reduced expression of these genes, demonstrating that these loci are related to transcriptional regulation of *KBTBD6* and *POP5*. Our results demonstrate the utility of telomere length GWAS in the identification of telomere length regulation mechanisms and validate *KBTBD6* and *POP5* as genes affecting telomere length regulation.

Telomeres shorten with age and short telomeres are associated with several age-related diseases including bone marrow failure and immunodeficiency[1]. Individuals with these short telomere syndromes have rare variants with large effects on telomere length regulation genes. Identification of causal variants in short telomere syndrome patients has led to the discovery of several genes we now appreciate as core telomere length regulation genes including *DKC1*, *NAF1*, *PARN*, and *ZCCHC8*[2–4]. Rare and common variants highlight the same set of core genes for many complex traits[5], therefore a genome-wide association study (GWAS) on telomere length could feasibly be used to discover additional critical telomere length regulation genes. Despite the fact that 19 GWAS on leukocyte telomere length have been published[6–24], identifying 143 loci associated with telomere length, very little has been done to validate these signals representing facets of telomere length regulation.

A key challenge facing the interpretation of telomere length GWAS signals is accurately identifying causal genes driving the association signals. The vast majority of GWAS signals, including telomere length GWAS loci, are in non-coding regions, making it difficult to determine the likely causal gene. Some telomere length GWAS have used colocalization analysis, statistically comparing a GWAS signal to quantitative trait locus (QTL) data, to support a shared causal signal with putative target genes[22–24]. Each of these were limited to expression QTLs (eQTLs) highlighting transcriptional regulatory genetic effects, but additional mechanisms may be involved, including alternative splicing revealed by splicing QTLs (sQTLs). Furthermore, colocalization evidence does not confirm causal genes or relevant cell types. Such conclusions require functional validation of genetic regulatory effects and of gene impact on telomere length, which were not explored in prior telomere length GWAS.

A second barrier to capitalizing on telomere length GWAS-associated loci is that many of the associated loci are often in or near genes with no prior known direct effect on telomere length, making it difficult to understand the value in characterizing the underlying molecular mechanisms. Indeed, many of these association signals likely represent peripheral genes with indirect mechanisms on

✉e-mail: rmathias@jhmi.edu; ajbattle@jhu.edu

telomere length regulation. This is consistent with observations from screens assaying the effect of knock-out libraries in *Saccharomyces cerevisiae* (*S. cerevisiae*) on telomere length which identified genes involved in diverse pathways either lengthening or shortening telomeres[25]. Similarly, immunoprecipitation followed by mass spectrometry of *S. cerevisiae* telomerase components identified interactions with proteins that have diverse functions[26]. In both types of experiments, the majority of the results were interpreted as indirect mechanisms of telomere length regulation. However, validation of genes identified in these studies has also identified direct effects on telomerase[27].

Here, we leveraged four telomere length GWAS that used non-overlapping cohorts in a random-effects multi-ancestry meta-analysis on 211,369 individuals to identify 56 loci, five of which were novel, associated with human telomere length. Using stratified linkage disequilibrium score regression (S-LDSC)[28] and enrichment analysis of Roadmap Epigenomics chromatin data[29] we determined that blood and immune cells were the most relevant cell type for telomere length association signals. We validated some of our colocalization analysis results in cultured cells and demonstrated that overexpression of *KBTBD6* and *POP5* increased telomere length as predicted by our statistical analyses. CRISPR/Cas9 deletion of the predicted causal regions for signals attributed to these genes in immortalized blood cells reduced the expression of both genes, further supporting the conclusion that *KBTBD6* and *POP5* are the causal genes at these telomere length association signals. Together this work shows the utility of human telomere length GWAS in identifying aspects of telomere biology.

## Results

### Multi-ancestry meta-analysis of leukocyte telomere length identifies 5 novel signals

We leveraged four GWAS with non-overlapping cohorts in a multi-ancestry meta-analysis of 211,379 individuals. Three studies were homogenous ancestries of European[22], Singaporean Chinese[21], or Bangladeshi[19] individuals. The fourth study used HARE to broadly categorize individuals as European, African, Asian, or Hispanic/Latino and generated ancestry-specific summary statistics[24](Supplementary Data 1). We meta-analyzed these seven sets of summary statistics and broadly refer to the Asian, Singaporean Chinese, and Bangladeshi individuals as Asian in this manuscript (Fig. 1). Across the four studies telomere length was estimated from blood leukocytes computationally from whole genome sequencing data using TelSeq[24] or experimentally using qPCR or a Luminex-based platform[19,21,22]. These studies previously demonstrated that all three assays are well correlated with telomere Southern blots. We used a random-effects model to identify 56 genome-wide significant loci (*p*-value < 5 × 10⁻⁸) including five novel signals (Fig. 1, Supplementary Data 2, Methods). We identified lead SNPs at each meta-analysis association signal (Methods) and examined the impact of study heterogeneity using Cochran's *q* statistic and I² statistic on the lead SNPs (Methods). None of the lead SNPs had significant heterogeneity by either measure (Supplementary Data 2). Loci were considered novel if there were no other reported sentinels within one megabase of the lead single nucleotide polymorphism (SNP) at the locus. We were able to examine four of our novel signals for replication in an independent telomere length GWAS[23] (Methods) and determined that two were directionally consistent and had nominal evidence of replication (Supplementary Fig. 1, Supplementary Data 2). Further comparison of lead SNPs at the novel loci across individual GWAS used in this meta-analysis showed that these signals only reach genome-wide significance (*p*-value < 5 × 10⁻⁸) in the meta-analysis (Supplementary Data 3).

### Fine-mapping analyses nominate putative causal variants and genes affecting telomere length

We used colocalization analysis[30] to determine whether each of our GWAS signals overlapped a signal from an independent quantitative trait locus (QTL) dataset (Methods), indicating causal genetic variants

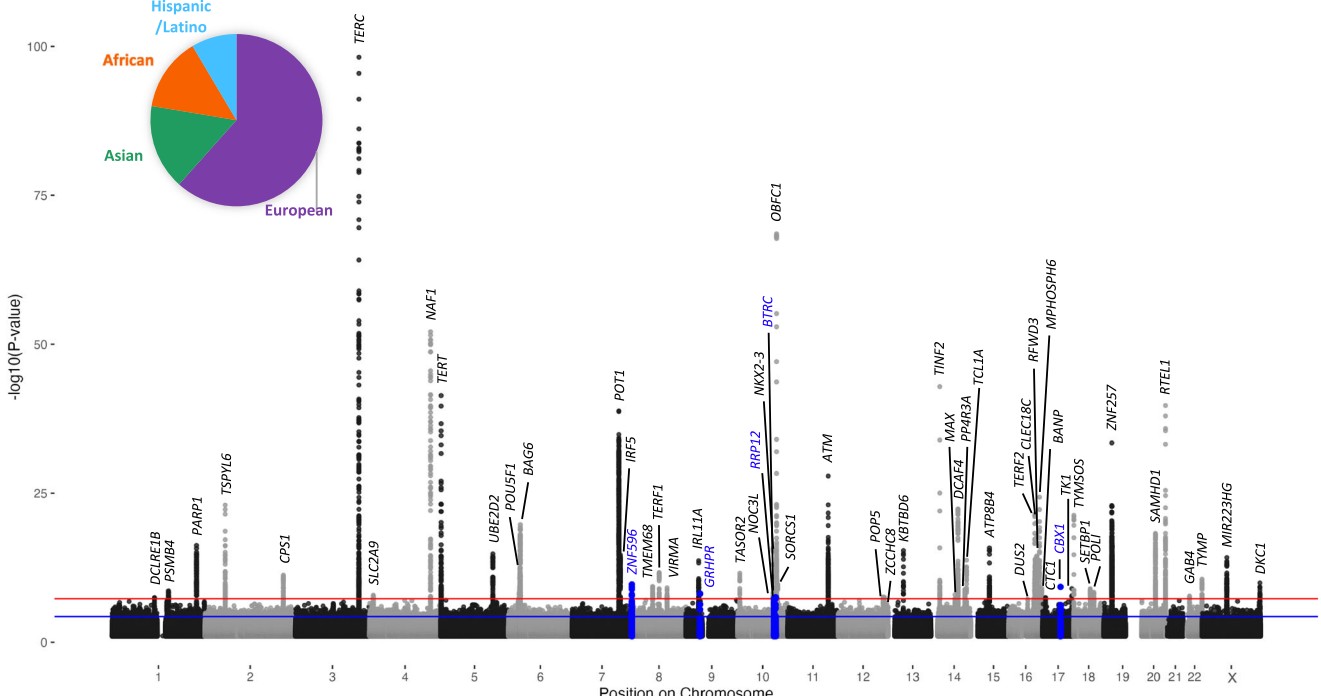

**Fig. 1 | Multi-ancestry meta-analysis of leukocyte telomere length identifies 5 novel signals.** Manhattan plot showing the results from the GWAS meta-analysis. SNPs with *p*-value < 0.1 are plotted. The novel signals are shown in blue. Red line indicates genome-wide significance after multiple testing correction (*p* < 5 × 10⁻⁸). The blue horizontal line indicates a suggestive threshold (*p* < 5 × 10⁻⁵). *N* = 211,369 individuals. The inset pie chart displays the proportion of broad ancestry groups used in the meta-analysis (Supplementary Data 1 and Source Data are provided as a Source Data file).

shared between telomere length and gene regulation. We began by examining large-scale expression quantitative trait locus (eQTL) and splicing quantitative trait locus (sQTL) datasets from diverse cellular contexts from the GTEx consortium. Each GWAS included in our meta-analysis estimated telomere length from leukocytes extracted from whole blood. However, strong QTLs are often shared across cellular contexts[31] and telomere length is correlated across GTEx tissues[32]; therefore, we included all 49 GTEx v8 tissues in our colocalization analysis. We found that 31 of 56 meta-analysis signals strongly colocalized (PPH4 > 0.7) with at least one eQTL or sQTL in at least one tissue (Supplementary Fig. 2A, B, E). 14 signals colocalized with an eQTL or sQTL across more than five tissues and there was colocalization of at least one meta-analysis signal with at least one eQTL or sQTL in 45 out of 49 GTEx tissues (Supplementary Data 4–5). We also conducted colocalization analysis using eQTLGen eQTLs[33] and DICE eQTLs[34] (Supplementary Data 6–7). eQTLGen has increased power, with 31,685 individuals compared to GTEx whole blood with 755 individuals. DICE introduced cell type specificity, with eQTLs called from RNA-seq on 13 sorted blood and immune cell types in 91 individuals. 11 of our signals colocalized with eQTLGen eQTLs (Supplementary Fig. 2C) and 9 signals colocalized with DICE eQTLs in at least one cell type (Supplementary Fig. 2D). Together, we found colocalization data to suggest putative target genes for 33 of our 56 signals (Fig. 2A). Two signals colocalized in all four QTL datasets and four signals colocalized with a GTEx eQTL, a GTEx sQTL, and an eQTLGen eQTL. 17 signals only colocalized in one dataset (Fig. 2B).

Next, we employed a transcriptome-wide association study (TWAS) approach, an alternative to colocalization analysis for nominating putative causal genes underlying meta-analysis signals. Often TWAS and colocalization analysis produce discordant results and it has been suggested that they capture complementary aspects of complex trait biology[35]. TWAS leverages expression data from a reference source to impute expression levels over a reference linkage disequilibrium dataset to generate predicted expression of genes which can be tested for association with summary statistics from a GWAS or meta-analysis. We used a pre-trained predicted expression model based on 1264 whole blood samples from the Young Finns Study[36] and used FUSION[37] to conduct the TWAS (Methods). We observed 19 significant results, nine of which were not located within 100 megabases of a genome-wide significant meta-analysis signal. This demonstrates the increased power of TWAS to detect genes significantly associated with a trait, as opposed to eQTLs which detect SNPs. Of the ten TWAS significant results proximal to genome-wide significant meta-analysis signals, seven agreed with genes nominated colocalization analysis (Supplementary Note 1). In the case of the meta-analysis signal led by rs7923385, a novel telomere length association signal, TWAS nominated *RRP12* as a putative causal gene while there was no supporting colocalization analysis data for that meta-analysis signal.

To identify putative molecular mechanisms underlying each signal, we synthesized the available data to converge on a high likelihood candidate gene, where possible (Methods, Supplementary Note 1). 28 meta-analysis signals colocalized with QTLs for one gene but in multiple cellular contexts (Supplementary Data 4–5). For example, the signal led by rs10111287 colocalized best with a *VIRMA* eQTL in thyroid (Fig. 2C), but also significantly colocalized with *VIRMA* eQTLs in stomach and whole blood. This signal only significantly colocalized with *VIRMA* eQTLs which made it straightforward to conclude this signal is likely linked to regulating *VIRMA* gene expression. Importantly, these results are not sufficient to make conclusions about the relevance of specific cellular contexts. Observed colocalization tends to correlate with the strength of the QTL, exemplified by the trend across the *VIRMA* eQTLs in thyroid (eQTL min $p = 3.79 \times 10^{-9}$, PPH4 = 0.922), stomach (eQTL min $p = 5.94 \times 10^{-7}$, PPH4 = 0.758), and whole blood (eQTL min $p = 2.13 \times 10^{-5}$, PPH4 = 0.567). Variable power in eQTL data

across tissues or cohorts is one reason that colocalization analysis is limited to suggesting candidate causal genes but not relevant cellular contexts[38].

13 meta-analysis signals colocalized (PPH4 > 0.7) with a GTEx sQTL (Fig. 2A, B), of which 6 also colocalized with an eQTL for the same gene (Supplementary Fig. 2E). sQTLs are called based on exon read depth relative to other exons in the splicing cluster; a reduction in the expression levels of just one exon can result in the locus also being reported as an eQTL due to fewer total reads mapping to the gene. Therefore, it is possible for a signal regulating splicing to have colocalization results with an sQTL and an eQTL. This was the case for the signal led by rs7193541 (Fig. 2D) which colocalized with an *RFWD3* sQTL in cultured fibroblasts (PPH4 = 1.000) and an *RFWD3* eQTL in skeletal muscle (Supplementary Note 1, PPH4 = 0.993). This meta-analysis signal also colocalized with an *RFWD3* sQTL in two other GTEx tissues (EBV-transformed lymphocytes and brain cerebellar hemisphere) and an *RFWD3* eQTL in seven other GTEx tissues (adipose visceral omentum, adrenal gland, breast mammary tissue, liver, prostate, minor salivary gland, and transverse colon). We can be confident that splicing is the likely molecular mechanism if the splicing cluster is clear and supported by effects on expression over affected exons. A LeafCutter plot of this splicing cluster demonstrated that individuals heterozygous (T/C) or homozygous (C/C) for the alternate allele at this locus increasingly excluded the fourteenth exon in *RFWD3* (Fig. 2D). This was further supported by examining the RNA expression alignment which showed decreased expression of only the fourteenth exon in individuals heterozygous (T/C) or homozygous (C/C) for the alternate allele (Supplementary Fig. 2F). This exon is excluded in observed RFWD3 protein isoforms (NP_001357465.1). These results lend strong support to the conclusion that this meta-analysis signal is driven by the association of telomere length with the regulation of *RFWD3* splicing and is it possible that this isoform may have distinct molecular effects on telomere length.

While colocalization analysis is an excellent tool for identifying potential causal genes for a meta-analysis signal, comparison across diverse cellular contexts and between datasets at times led to multiple putative causal genes. There were 16 meta-analysis signal-gene QTL colocalization pairs that were replicated between datasets (Supplementary Fig. 2E). In 17 cases there was only colocalization evidence from one QTL dataset (Fig. 2B) and in 12 cases there was conflicting colocalization results for a meta-analysis signal (Supplementary Note 1). For example, the signal led by rs59922886 colocalized strongly with a *CTC1* eQTL in GTEx sun exposed skin (PPH4 = 0.861). But in eQTLGen the same meta-analysis signal best colocalized with an *AURKB* eQTL (PPH4 = 0.919). Colocalization analysis from DICE further supported attribution to *CTC1* where the signal colocalized with a *CTC1* eQTL in M2 cells (PPH4 = 0.641). In this case, known biology allowed us to confidently attribute the signal to *CTC1* because CTC1 functions as part of the CST complex to regulate telomere length.

Recently there has been discussion about whether assigning genes to GWAS or meta-analysis signals should rely upon colocalization analysis as opposed to the proximal gene[39]. 18 of our 56 meta-analysis signals best colocalized with the proximal gene and TWAS nominated the proximal gene at five of our 56 meta-analysis signals. We assigned a gene to each meta-analysis signal based on known biology of proximal genes, colocalization analysis results, or the proximal gene where no other information was available. We discuss these situations and our rationale for putative causal gene assignment in the Supplementary Note 1.

To identify putative causal SNPs at each locus we applied fine-mapping using SuSiE[40] to estimate 95% credible sets. This analysis results in a set of SNPs estimated to contain a casual SNP with 95% confidence based on GWAS summary statistics and linkage disequilibrium estimates. We were able to identify 95% credible sets at 38 of 56 loci (Supplementary Data 8, Methods). Those without a SuSiE

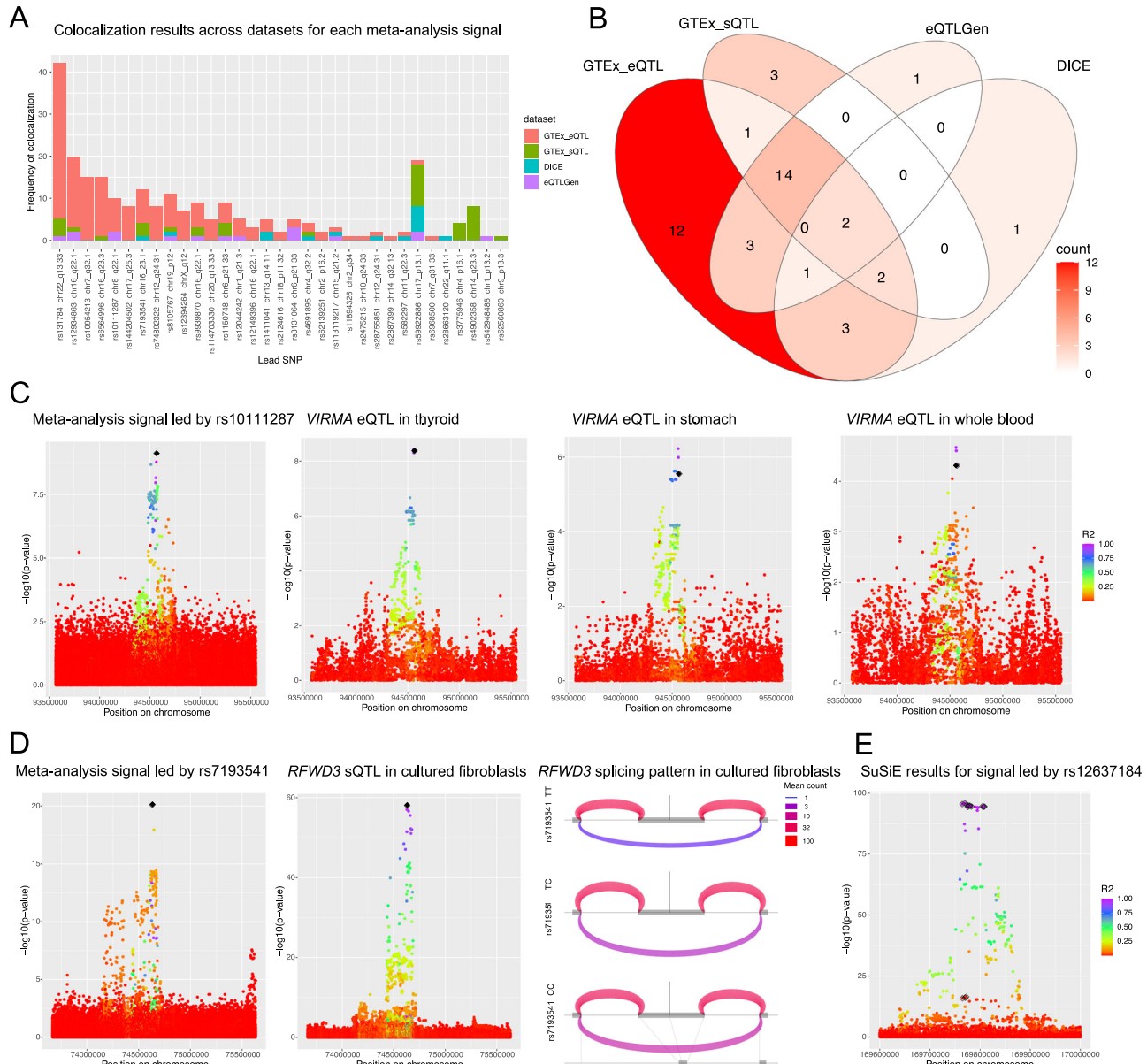

**Fig. 2 | Fine-mapping analyses nominate putative causal variants and genes affecting telomere length. A** Colocalization events between a meta-analysis signal and a QTL for any gene across QTL datasets. **B** Colocalization of meta-analysis signals with any gene QTL in any cell type. **C** Manhattan plots for the meta-analysis signal near rs10111287 colored by $r^2$ with the lead SNP (black diamond) and *VIRMA* eQTLs in three GTEx tissues. Colocalization results for each eQTL with the meta-analysis signal are indicated in the top right corner. Colocalization analysis between the eQTLs suggests there are shared causal SNPs: thyroid eQTL with stomach eQTL PPH3 = 0.090 PPH4 = 0.906, thyroid eQTL with whole blood eQTL PPH3 = 0.144 PPH4 = 0.745, stomach eQTL with whole blood eQTL PPH3 = 0.190 PPH4 = 0.655. **D** Manhattan plot for the meta-analysis signal near rs7193541 colored by $r^2$ with the lead SNP (black diamond) and *RFWD3* sQTL. Colocalization results for the QTL with

the meta-analysis signal are in the top right corner. In the LeafCutter splicing cluster diagram gray boxes represent the *RFWD3* exons involved in the splicing cluster, the central exon is exon 14 (hg38: chr16:74630780-74630957). Curved lines represent the average number of reads spanning each exon-exon junction across individuals. Thinner, purple curves represent lower expressed exon-exon junctions and thicker, pink/red curves represent higher expressed exon-exon junctions. The vertical gray line indicates the location of the lead SNP. The line at the bottom shows the linear base pair position of each exon and intron depicted in the plots. TT $N = 167$, TC $N = 236$, and CC $N = 80$. **E.** Manhattan plot showing the SuSiE 95% credible sets for the signal led by rs12637184. Credible set 1 (black diamonds, 10 SNPs) and credible set 2 (black squares, 4 SNPs). $r^2$ is calculated with respect to the lead SNP. Source data are provided as a Source Data file.

predicted credible set had weaker association compared to those with credible sets.

SuSiE identified two independent causal variants for the signal led by rs35510081 (Fig. 2E). We did not observe any significant colocalization results for this locus. It is not unusual for a considerable proportion of GWAS signals to not colocalize with QTLs and this may be due to the gene being under extreme selective pressure or having low expression[39,41–43]. In such cases, prior knowledge and proximity to nearby genes are commonly considered. In this case,

*TERC*, the RNA component of telomerase, is not the immediate proximal gene but is nearby (4.5 kilobases). Previous telomere length GWAS have attributed signals to *TERC* led by nearby SNPs rs2293607, rs12696304, and rs12638862[7,10,19,21,24]. Furthermore, given the information we have about *TERC* as a component of telomerase, we can be confident attributing this signal to *TERC*. In this and similar cases known biological information superseded the proximal gene or colocalization analysis results in assigning the peak (Supplementary Note 1).

16 of the 38 loci where credible set estimation was possible were predicted to have multiple causal SNPs. The number of predicted causal SNPs at each locus is consistent with previously published conditional analysis on the pooled ancestry GWAS[24] (Supplementary Fig. 2G). Many of these signals also have a stronger association with telomere length and the detection of multiple causal SNPs is likely due to increased power. The exceptions to this trend are the *TERF1* locus, which is a telomere-binding protein, and the *DCLRE1B* (aka *APOLLO*) locus, which is important for telomere end processing. The association signals at these loci were not as strong ($p = 2.04 \times 10^{-12}$ and $p = 3.26 \times 10^{-8}$, respectively) yet are estimated to have 6 and 3 causal SNPs at the signals, respectively. We previously demonstrated that the multiple signals at the *OBFC1* (aka *STN1*) locus colocalize strongly with *OBFC1* eQTLs in distinct tissues[24]. This is also true for *NAF1* (Supplementary Fig. 2H). Both *NAF1* and *OBFC1* could be considered core telomere length regulation genes as they have direct mechanisms on biosynthesis and regulation of telomerase and their independent signals could reflect distinct regulatory mechanisms across cellular contexts. However, as discussed above, QTL detection can be influenced by technical factors, and from this work alone we are unable to eliminate the possibility that there may be undetected QTLs in these cellular contexts that would colocalize with one another. But the prevalence of multiple causal SNPs at many association signals reiterates the importance of these core genes in telomere length regulation across cellular contexts.

## Genes suggested by colocalization analysis highlight nucleotide synthesis and ubiquitination

We looked for GO biological process pathway enrichment using PANTHER[44] and observed very strong enrichment of telomere regulation and DNA damage response pathways, as expected (Supplementary Data 9). We observed similar GO process enrichment using proximal genes and colocalization analysis-supported genes (Supplementary Fig. 3). We also observed significant enrichment of nucleotide synthesis processes (e.g. cellular aromatic compound metabolic process, nucleic acid metabolic process). The importance of dNTP pools in regulating telomerase has been well documented[45] and one of the GWAS included in our meta-analysis also highlighted the importance of nucleotide metabolism in telomere length regulation[22]. Though we did not observe enrichment of any protein degradation biological processes, we attributed several of our meta-analysis signals to genes involved in proteasomal degradation including *UBE2D2*, *KBTBD6*, *PSMB4*, and *RFWD3*. *UBE2D2* is proximal to the rs56099285 signal and is an E2 ubiquitin conjugating enzyme. The signal near rs1411041 colocalized strongly with both *KBTBD6* and *KBTBD7*; these neighboring genes function as part of an E3-ubiquitin ligase complex. Additionally, we observed a signal near rs12044242 which we attributed to *PSMB4*, a non-catalytic component of the 20S proteasome, and a signal near rs7193541 which we and others attributed to *RFWD3*, an E3 ubiquitin ligase. Together this collection of genes highlights an unappreciated role of ubiquitination regulation in telomere length regulation dynamics.

## Meta-analysis signals are enriched for transcription factor binding sites of transcription factors with roles in telomere length regulation

Several transcription factors are known to regulate core telomere genes and disruption or creation of their transcription factor binding sites can result in dysregulation of telomerase and telomere length regulation[46]. We examined whether the 95% credible set SNPs for our meta-analysis signals were enriched for transcription factor binding sites of any transcription factors with known consensus sequence using ENCODE ChIP-seq data (Fig. 3A)[47] or ReMap consensus sequences (Supplementary Fig. 4A, Methods)[48]. Some loci have larger 95% credible sets or multiple causal variants (Supplementary Data 8,

Supplementary Fig. 2G), while this analysis tests enrichment against a set of control SNPs that were matched based on the number of variants in linkage disequilibrium, minor allele frequency, and distance to the nearest gene of the index SNPs, it is possible that some loci had a stronger influence on enrichment. Therefore, we also analyzed the enrichment of the lead SNP alone at each meta-analysis signal (Supplementary Fig. 4B, C). Many transcription factors involved in telomere length regulation had binding sites that were enriched in our meta-analysis using both analyses (Fig. 3A, Supplementary Fig. 4A–C, Supplementary Data 10). The transcription factor binding site enrichment calculated using ENCODE data was correlated with that of ReMap (95% credible set analysis $R^2 = 0.336$, lead SNP analysis $R^2 = 0.589$)(Supplementary Fig. 4D–E).

Previous work demonstrated that PAX5 increases *TERT* expression in B cells[49]. We observed that there is a PAX5 transcription factor binding site overlapping the signal led by rs12044242, which we assigned to *PSMB4* (Supplementary Note 1). This SNP alters a highly weighted cytosine in the consensus sequence to a thymine and overlaps ChIP-seq peaks for activating histone marks (H3K4me3, H3K1me1, H3K27ac) and binding sites for transcriptional regulators (POL2, CTCF, HDAC1, HDAC2) (Fig. 3B). Lead SNPs at signals we attributed to *OBFC1* and *TINF2*, both of which produce key telomere binding proteins, overlap binding sites for SOX2 and KLF4, respectively. In addition, one of our novel signals, which we attributed to *RRP12*, overlaps a MYC binding site. SOX2, KLF4, and MYC are pluripotency factors and the presence of their binding sites at these telomere length association signals suggests regulatory roles for these genes in pluripotent cells. Furthermore, MYC is a well-established regulator of *TERT* expression. Our meta-analysis lead SNPs also overlapped transcription factor binding sites for FOXE1, GABPA, and HMBOX1 (Supplementary Data 11) which have all been reported to regulate expression of *TERT*, the protein component of telomerase[50–52]. Present literature on this topic has been focused on transcription factors regulating telomerase; these results demonstrate that these transcription factors may regulate other key telomere length regulation genes. Furthermore, the strong enrichment of some transcription factors with no known role in regulating telomere length regulation genes (Fig. 3A) may direct future experiments toward transcription factors critical to telomere length regulation.

## *TCL1A* 95% credible set SNPs are more strongly associated with telomere length in older individuals

Understanding molecular mechanisms underlying GWAS signals is further complicated by temporal specificity; some genetic effects are stronger during specific stages of development or age[53]. Because age accounts for a significant amount of telomere length variation[32], we ran a GWAS with an interaction term between age and genotype. Five signals had a genotype x age p-value that was below genome-wide significance ($p$-value $< 5.39 \times 10^{-9}$) and another 48 signals had genotype × age $p$-values that were suggestive ($p$-value $< 5 \times 10^{-5}$) (Supplementary Data 12). None of the genome-wide significant interaction signals were within two megabases of a meta-analysis signal, therefore we ran a GWAS stratified by age as an orthogonal approach (Supplementary Data 13). This analysis required individual-level data and was therefore limited to the 109,122 individuals from TOPMed. We divided these individuals into three age groups ([0, 43], (43, 61], and (61, 98]) such that there were a similar number of individuals in all three groups. Expanding the analysis to more granular age groups was not possible with this sample size without singularity issues in the GWAS analysis. Although the ratio of males to females was similar between groups (Supplementary Fig. 5A), the distribution of ancestries varied such that the proportion of European individuals increased over age (Supplementary Fig. 5B).

Comparison of the GWAS results with and without the age and genotype interaction effect showed little overlap in signals that were

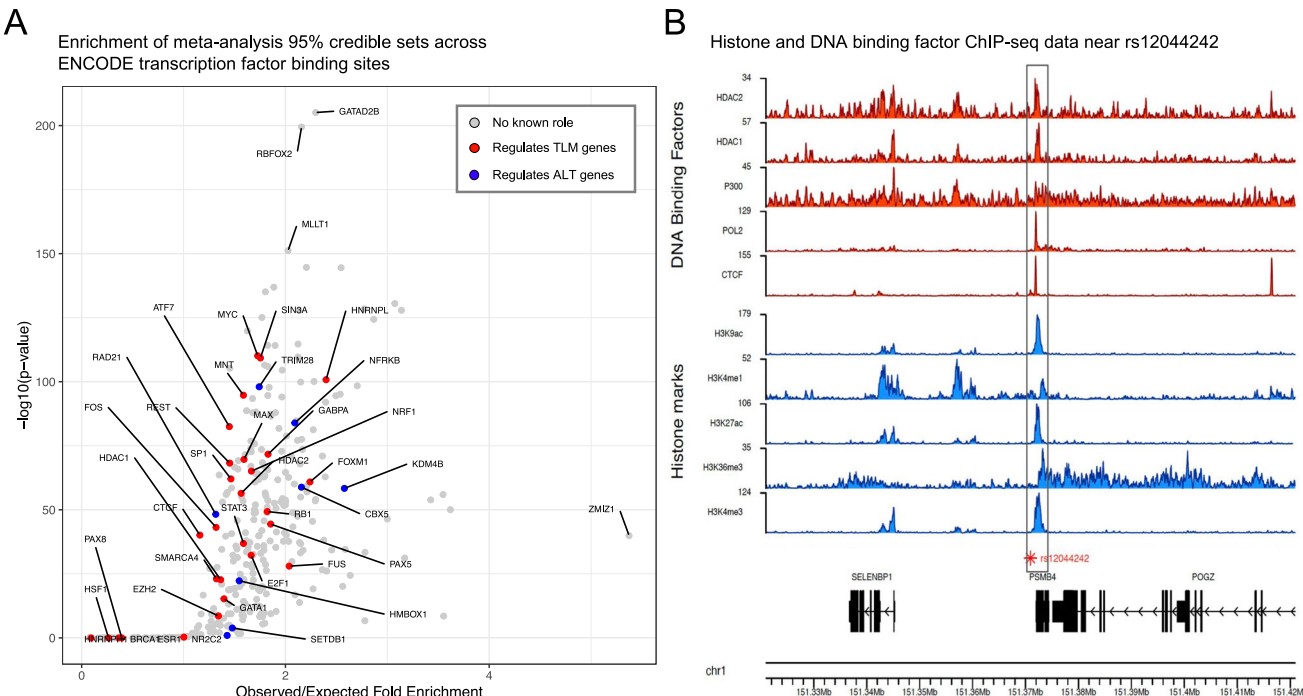

**Fig. 3 | Enrichment analysis of transcription factor binding sites of transcription factors with roles in telomere length regulation highlights a PAX5 binding site near *PSMB4*. A** Enrichment of 95% credible set SNPs across all transcription factors with ChIP-seq data available from ENCODE using a one-sided binomial test (Methods). Red points represent transcription factors with known roles in regulating telomere length maintainence (TLM) genes and blue points represent transcription factors with known roles in the alternative telomere lengthening (ALT) pathway. There were 320 transcription factors plotted (28 red, 8 blue, 284 gray). There were 18 transcription factors that fall at the (0,0) coordinate that are not plotted for the sake of clarity; one (XRCC3) had known roles in ALT. A complete list of transcription factors is provided in Supplementary Data 9 and source data are provided as a Source Data file. **B** ChIP-seq data for the indicated DNA binding factor (red) or histone mark (blue) was generated by ENCODE and downloaded as bigwig files from the UCSC genome browser. The gene structure and genomic coordinates are depicted below the ChIP-seq data.

significant in both analyses (Supplementary Fig. 5C). The GWAS without the age and genotype interaction term is a replicate of the previously reported TOPMed pooled GWAS[24]. There were three suggestive signals ($p < 5 \times 10^{-5}$) in the age and genotype effect GWAS that were also suggestive in the TOPMed pooled GWAS (Supplementary Fig. 5D). The signal led by rs2515349 was a single associated SNP with no supporting association peak despite linkage disequilibrium in the region (Supplementary Fig. 5E), therefore we did not examine it further. The signal led by rs585168 was suggestive in the GWAS with an age and genotype interaction term and part of an association signal both the TOPMed pooled GWAS and our meta-analysis (Supplementary Fig. 5F). In the meta-analysis we attribute this signal to *MIR223HG*. However, the effect size estimate of rs585168 did not have an apparent linear effect with age in the age-stratified GWAS (Supplementary Fig. 5G). Further examination of this locus in a larger cohort may further elucidate the relationship between this locus and age. The third signal, led by rs2296312, was part of an association signal in the TOPMed pooled GWAS (Fig. 4A), was part of a suggestive association signal in the genotype and age GWAS (Fig. 4B), and is part of an association signal in our meta-analysis (Supplementary Note 1). The effect size estimate of rs2296312 increased over age (Fig. 4C) and this trend was independent of ancestry as the effect estimate for rs2296312 was similar between all examined ancestries (Fig. 4D). The association signal increased in significance over age in our age-stratified GWAS, mirroring the effect size estimate trend over age (Fig. 4E). In the meta-analysis, rs2296312 was part of a peak that colocalized best with a *TCL1A* eQTL from GTEx whole blood (PPH4 = 0.714). SuSiE credible set analysis identified 14 SNPs in the credible set for this peak all of which have a similar trend in their effect estimates over age (Supplemental Data 13). Together these data demonstrate that putative causal SNPs

regulating *TCL1A* expression are associated with age and telomere length. TCL1A activates the AKT signaling pathway increasing cellular proliferation[54] and *TCL1A* expression was previously reported to decrease in whole blood as age increases[32]. Furthermore, rs2296312 has been reported to act through *TCL1A* to be protective against loss of the Y chromosome and clonal hematopoesis[55,56]. Our data are concordant with previous findings and suggest that these protective phenomena reduce proliferation, leading to longer telomere length.

## Blood and immune cells are a key cell type for leukocyte telomere length

To enable experimental validation of putative causal genes underlying our meta-analysis signals, we first had to identify cellular contexts in which the majority of our association signals were relevant. Telomerase is active in stem and progenitor cells in addition to peripheral blood leukocytes and bone marrow; however, telomere length regulation is relevant in many different cell types. In relevant cellular contexts, causal SNPs are expected to be in genomic regions with active chromatin states. We tested for enrichment of the meta-analysis lead SNPs across Roadmap Epigenomics samples (Supplementary Data 14) and the 25 state chromHMM model (Fig. 5A)[29]. We note that this dataset consists largely of terminally differentiated cell types in which telomerase is not active, however, the telomerase components (*TERT* and *TERC*) are only two of our 56 association signals and we expect that the majority of our association signals represent telomere length regulation mechanisms that are active across many cell types. We identified the cell type group with the strongest enrichment for each chromatin state (Methods) and observed that the blood and T-cell cell type group had the strongest enrichment across the most active chromatin states (Supplementary Fig. 6A). The chromHMM

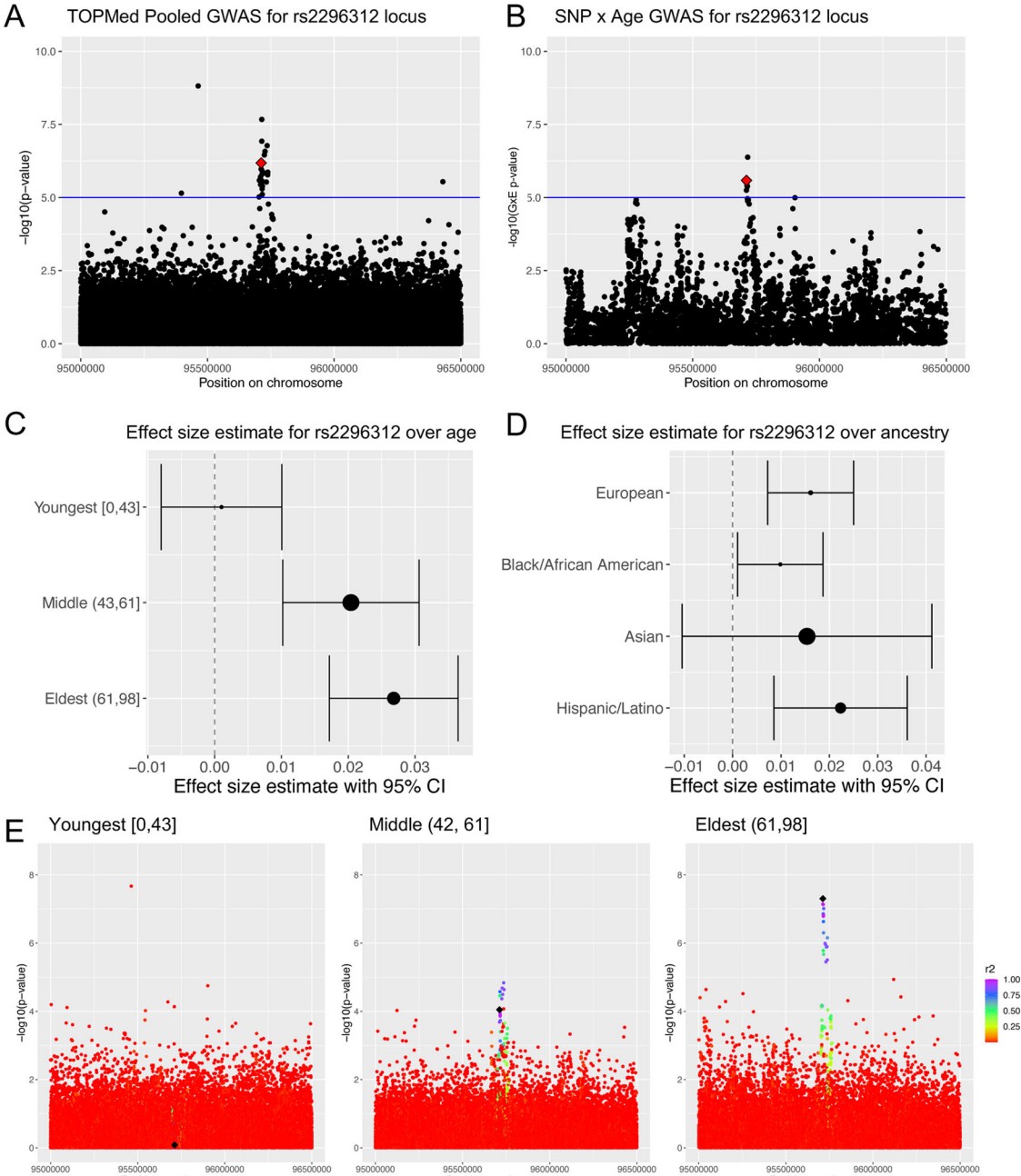

**Fig. 4 | *TCL1A* 95% credible set SNPs are more strongly associated with telomere length in older individuals. A**, **B** Manhattan plot for the region around rs2296312 (red diamond) using (**A**) summary statistics from TOPMed Pooled GWAS (**B**) summary statistics from age and genotype interaction GWAS. The log10(*p*-value) for the interaction covariate is plotted on the *y*-axis. **C**, **D** 95% confidence interval for the effect size estimate is shown and the size of the data point reflects the standard error. **C** Effect size estimate for rs2296312 (tested, minor allele = C) across

age groups from the age-stratified GWAS. **D** Effect size estimate for rs2296312 across ancestry groups from ancestry-stratified GWAS[24]. European MAC = 16,443; Black/ African MAC = 19,963; Asian MAC = 5,683; Hispanic/Latino MAC = 18,019. **E** Manhattan plots for the rs2296312 (black diamond) locus in age-stratified GWAS. Color indicates *r*² calculated with respect to rs2296312. Source data for Fig. 4 are provided as a Source Data file.

model was trained on ChIP-seq data from five core histone epigenetic marks and generated a genome-wide, tissue specific predicted chromatin state[29]. Because the chromHMM model is a predicted state, we also examined whether there was enrichment when looking at the primary ChIP-seq data for two of the core histone epigenetic marks. Consistent with the chromHMM model results, we saw that the strongest enrichment of lead SNPs in H3K4me1 and H3K27ac peaks was in blood and T-cell samples (Supplementary Fig. 6B, C).

As an orthogonal approach we ran stratified linkage disequilibrium score regression (S-LDSC) on the meta-analyzed European

individuals in our study (Methods). S-LDSC uses the meta-analysis summary statistics to examine whether, given linkage disequilibrium, a category of SNPs has increased association with telomere length compared to SNPs not in that category. In this case, we used categories previously reported for cell type specific annotations based on gene expression or chromatin marks[28]. As with the Roadmap Epigenomics data, the cell types analyzed in these datasets are largely terminally differentiated, though progenitor and stem cells were included. Using both gene expression and chromatin marks we observed that the blood/immune cell tissue category was the only tissue category that

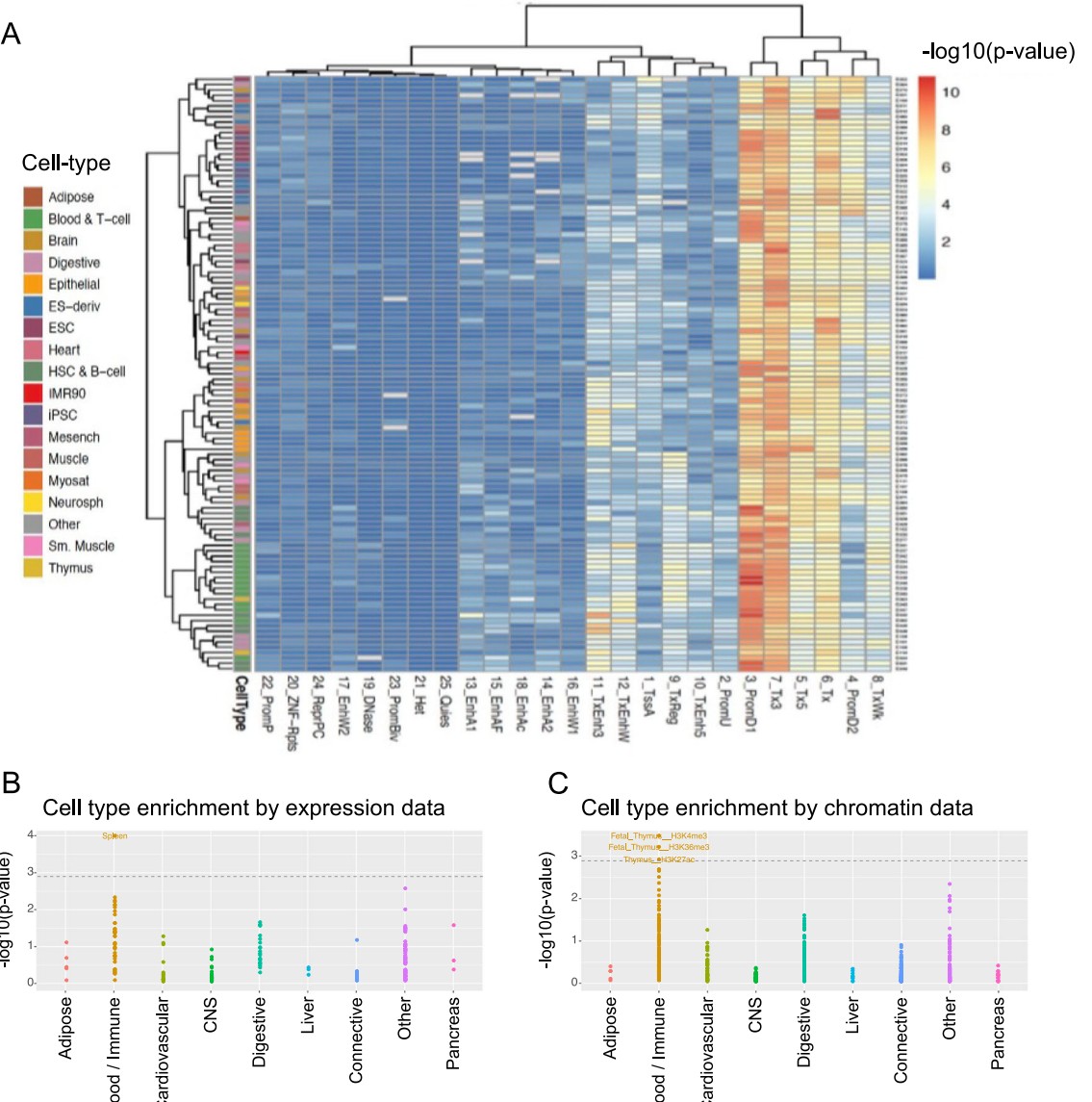

**Fig. 5 | Blood and immune cells are a key cell type for telomere length.**
**A** Hierarchical clustering of the enrichment of meta-analysis lead SNPs in predicted states using the Roadmap Epigenomics 25 state chromHMM model (*p*-values from a one-sided binomial test). Dark red cells indicate the strongest enrichment, largely in predicted state 3: PromD1 (Promoter Downstream TSS 1) and largely for rows corresponding to Blood and T-cell samples. **B**, **C** Stratified LDSC was conducted on

130,246 meta-analyzed European individuals [22,24] using the 1000 Genomes European linkage disequilibrium reference panel. Each dot represents a cell type assigned to the broader tissue categories specified on the x-axis by Finucane et al. 2018. The gray dotted line represents the significance threshold of FDR < 0.05 at −log10(*p*-value) = 2.75. Source data for Fig. 5 are provided as a Source Data file.

was significantly enriched (Fig. 5B, C). Together with the Roadmap Epigenomics enrichment analysis, these data suggest that blood and immune cells are the most relevant cell type for genetic regulation of leukocyte telomere length (Discussion).

**Overexpression of *POP5* and *KBTBD6* increases telomere length in HeLa-FRT cells**

We began our validation experiments by screening candidate genes for an effect on telomere length. It has been well documented that shRNAs with loss of function effects often become epigenetically silenced over time in cell culture. Therefore, we limited our candidate genes to those that were predicted to affect telomere length when their expression was increased. We identified genes with eQTLs in any GTEx tissue that colocalized with our meta-analysis signals at a reduced threshold of PPH4 > 0.5. Next we required that the tested allele of the lead SNP at the meta-analysis signal also have a significant effect (FDR < 0.05 in

GTEx) on the expression of the candidate gene and be associated with increased gene expression as the tested allele copy number increased in GTEx (Methods, Supplementary Note 2). Of those we chose five genes that had one known protein coding sequence isoform, had strong colocalization analysis results, and had some known biology: *OBFC1*, *PSMB4*, *CBX1*, *KBTBD6*, and *POP5* (Methods, Supplementary Note 2). To generate constitutive overexpression cell lines we used the Flp-in system to incorporate the FLAG-tagged gene of interest under the control of a CMV promoter into HeLa-FRT cells (Methods). HeLa cells are not derived from blood or immune cells but are highly tractable for this screening stage of the validation experiments. Three independent transfection clones were passaged and the effect of gene overexpression on telomere length was observed by Southern blot.

The lead SNPs for each meta-analysis signal that we attributed to these genes was estimated to have a positive effect on telomere length in our meta-analysis (Supplementary Data 2), therefore we predicted

that overexpression of these genes should increase telomere length. As a control we also overexpressed *GFP*, which had no effect on telomere length, as expected (Fig. 6). Overexpression of *OBFC1* or *PSMB4* also had no effect on telomere length (Supplementary Fig. 7A). Overexpression of *CBX1* slightly increased telomere length (Supplementary Fig. 7A). Overexpression of *KBTBD6* increased telomere length over time in clone 5 while telomere lengthening plateaued in clone 7 (Fig. 6). Overexpression of *POP5* increased telomere length in both clone 5 and clone 6 initially but then lengthening plateaued (Fig. 6). The median, minimum, and maximum telomere lengths were estimated for each lane in the Southern blots using ImageQuant TL (Methods, Supplementary Fig. 8). Protein expression was assayed by western blot analysis. Western blot comparison of early population doubling timepoints to late population doubling timepoints showed that *POP5* overexpression was maintained through the duration of the experiment while *KBTBD6* overexpression was suppressed in clones 6 and 7 late timepoints (Supplementary Fig. 7B, C). This likely accounts for the plateau in telomere lengthening in *KBTBD6* overexpression clone 7 (Fig. 6A, B). Previous work has observed that when a telomere length regulation protein was overexpressed telomeres lengthened but plateaued over time and our experiments are consistent with this[57]. The overall increase in telomere length in response to *KBTBD6* or *POP5* overexpression is consistent with what was predicted by our computational analyses.

### CRISPR removal of *KBTBD6* and *POP5* regulatory regions reduced expression of each gene

We next sought to examine whether high likelihood causal elements in the respective meta-analysis signals affect the expression of these genes. SuSiE was unable to predict a 95% credible set for the *POP5* locus, likely because the association signal is below genome-wide significance in the summary statistics used for fine-mapping (Methods). We utilized a second credible set estimation algorithm, CAVIAR[58], with a single assumed causal SNP, however, the 95% credible set included 3041 SNPs and did not reduce the position range of the region (Supplementary Fig. 9A). In the absence of useful 95% credible set estimation, we considered the genome region spanning the lead SNP and SNPs with $r^2 > 0.9$ and $p$-value $< 1 \times 10^{-6}$ (Supplementary Fig. 9B). To prioritize a subset of this 124 kilobase region, we intersected these top SNPs with ATAC-seq, Hi-C, and chromatin ChIP-seq data from blood samples, but were unable to form a consensus. We removed the 124 kilobase region upstream of *POP5* using CRISPR/Cas9 in K562 cells (Supplementary Fig. 9C) and identified 24 clones where the region had been successfully deleted at one allele, generating heterozygous deletions (Methods). qPCR analysis (primer sequences in Supplementary Data 15) of these clones showed significantly reduced *POP5* expression compared to controls ($p = 0.047$) demonstrating that this region contains critical SNPs for regulating *POP5* expression in blood cells (Fig. 7A).

KBTBD6 functions as a component of an E3 ubiquitin ligase complex along with CUL3 and KBTBD7. *KBTBD7* is a neighboring gene and we observed colocalization with the signal led by rs1411041 with both *KBTBD6* and *KBTBD7* eQTLs in GTEx (Supplementary Data 4). We were interested in determining whether CRISPR editing of high likelihood SNPs in this meta-analysis signal would affect the expression of *KBTBD6*, *KBTBD7*, or both. We intersected the position of the 99% credible set SNPs (Fig. 7B) with ATAC-seq peaks in blood samples (Fig. 7C). Only one SNP, rs9525462, was located in a region where the ATAC-seq peaks were shared across blood samples. rs9525462 was predicted to be in the 99% credible set by both SuSiE and CAVIAR. This region overlaps promoter and enhancer chromatin marks (H3K27ac and H3K4me3, respectively) in Roadmap Epigenomics blood samples (Fig. 7D), further supporting that this region is in an active state in blood samples. We used CRISPR/Cas9 to remove the 938 bp ATAC-seq peak region in K562 cells (Supplementary Fig. 9D) and identified 31

clones where this region had been successfully removed, generating heterozygous deletions (Methods). Clones with the ATAC-seq peak region knocked-out had significantly decreased *KBTBD6* ($p = 0.003037$) and *KBTBD7* ($p = 2.093e-05$) expression relative to controls, demonstrating that this region is critical in regulating the expression of both genes. Together these data demonstrate that our meta-analysis signals are driven by *POP5* and *KBTBD6/KBTBD7*, and we identify them as telomere length regulation genes.

## Discussion

Our results demonstrate the utility of telomere length GWAS in the identification of telomere length regulatory mechanisms. Our fine-mapping of telomere length-associated loci and discussion of relevant cell types in which to validate these signals is a useful platform for further experimental validation. We determined that blood and immune cells are the most relevant cellular context to examine leukocyte telomere length association signals based on chromatin accessibility and S-LDSC. This was not a surprising result as telomere length was estimated from blood leukocytes in all samples and it is possible that this boosted the strength of blood and immune cell enrichment in our analyses. However, telomere length regulation is relevant in many different cell types, to differing extents, and it is possible that with a higher powered GWAS or with additional cell types that are currently underrepresented in the S-LDSC analysis, significant enrichment of additional cell type groups would be detectable. We propose that blood and immune cells are the most relevant cell type for leukocyte telomere length GWAS validation experiments, but that genes underlying leukocyte telomere length association signals contribute to telomere length regulation across cellular contexts. This idea is further supported by our observation that independent association signals at the *OBFC1*[24] and *NAF1* loci colocalize with eQTLs for their respective genes in different cellular contexts.

While prior telomere length GWAS[22,23] have used colocalization to support putative causal genes for their association signals, we extended this work to include multiple expression QTL datasets, splicing QTLs, and TWAS analysis. This made it possible to uncover splicing mechanisms that may be associated with telomere length, as we saw with *RFWD3*, and increased the confidence of our putative causal gene assignment. TWAS analysis supported the association of *RRP12* with telomere length, which is proximal to one of our telomere length association signals. In addition, the TWAS analysis supported six genes nominated by colocalization analysis, demonstrating the value of applying diverse methods to nominate putative causal genes underlying meta-analysis association signals. Our SuSiE 95% credible set estimation suggested there were multiple, independent causal variants at sixteen loci, however, coloc assumes a single causal variant. Future work using methods such as CAFEH[38], which can identify shared causal variants across datasets, will be valuable for further investigating the non-primary signals at these loci.

Fifteen genes have been implicated in short telomere syndromes[1,4,59]; however, a significant proportion of short telomere syndrome patients lack a genetic diagnosis. Nine of these genes have also been identified as associated with telomere length based on GWAS using common genetic variants, consistent with observations that rare and common variants highlight the same set of core genes for many complex traits[5]. This suggests that genes nominated by leukocyte telomere length GWAS may point to additional candidate genes for short telomere syndrome patients lacking genetic diagnoses.

Experimental validation of putative causal genes identified genes involved in telomere length regulation. POP5 is a subunit of the Ribonuclease P/MRP complex. Previous work in *S. cerevisiae* demonstrated a role for specific components of the homologous complex in telomerase holoenzyme complex regulation[27]. In addition, POP1, another subunit of the Ribonuclease P/MRP complex, was recently shown to interact with human telomerase RNA[60]. Together, these

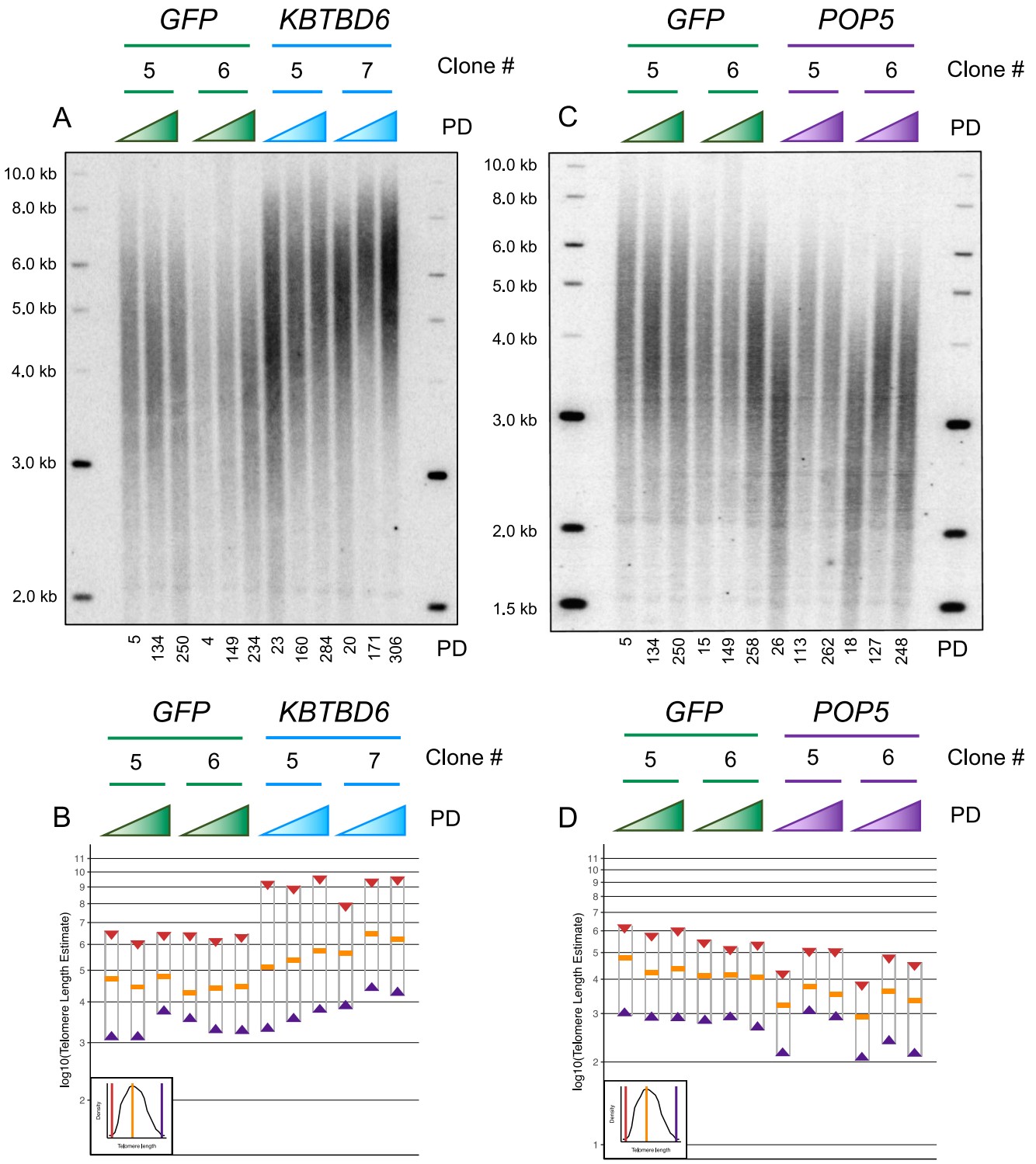

**Fig. 6 | Overexpression of *POP5* or *KBTBD6* increases telomere length in HeLa-FRT cells.** *KBTBD6*, *POP5*, or *GFP* was constitutively overexpressed from the CMV promoter in HeLa-FRT cells using the FLP-in system. **A**, **C** Telomere Southern blots showing the bulk telomere length from a population of cells. Molecular weight standards were run alongside the samples and their size is indicated in kilobases (kb). Three time points are shown for each clone and the estimated number of population doublings (PD) for each timepoint are indicated below the Southern blot. Each clone has the opportunity to form a distinct starting telomere length distribution which is why the first timepoint for some clones appear to have distinct telomere length distributions, for example the starting timepoint for the *POP5* clones compared to the *GFP* clones. All transfection experiments began from the same population of HeLa-FRT cells. Three biological replicates/clones for each overexpression gene were tested and the trends shown here were consistent across all clones in all cases. **B**, **D** The Southern blot densitometry was analyzed using ImageQuant TL to generate line plots of the pixel density. The software estimated the median telomere length (orange bar) as the pixels with greatest density and estimated a molecular weight for that position taking into account the molecular weight standards on both sides of the gel. The ImageQuant TL line plots (Supplementary Fig. 7) were used to estimate the minimum (purple triangle) and maximum (red triangle) telomere lengths in the bulk telomere band. A simulated diagram in the bottom left of the plot representing the ImageQuant TL plots is provided as a guide for the source of these values. The y-axis is plotted on a log10 scale to better estimate how linear DNA moves through an agarose gel at a rate inversely proportional to its length. Source data for Fig. 6 are provided as a Source Data file.

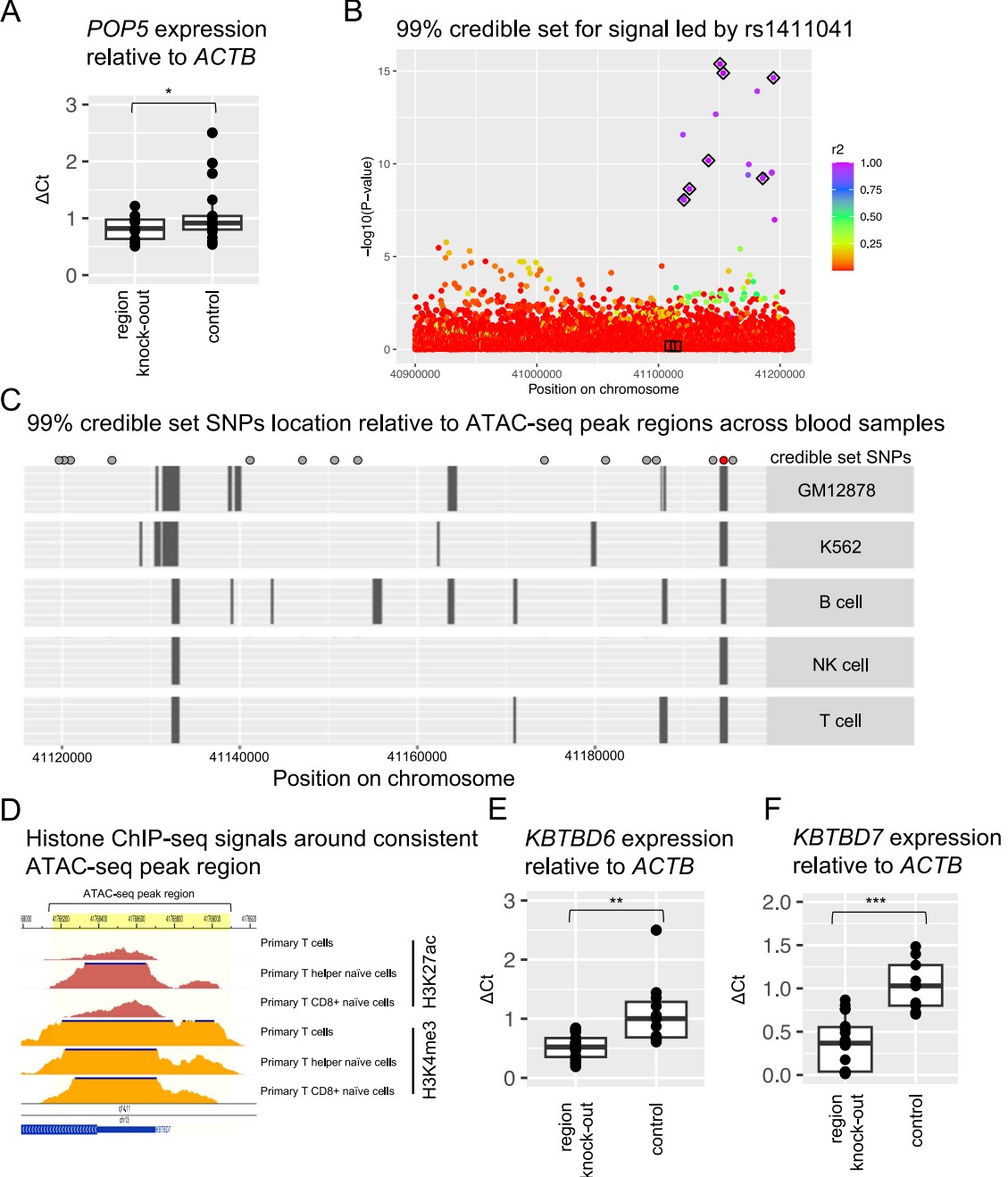

**Fig. 7 | CRISPR removal of *KBTBD6* and *POP5* regulatory regions reduced expression of each gene. A** Knock-out $N = 9$, control $N = 17$. One-sided $t$ test $p$-value = 0.047 (*). Boxplot center is the mean and box bounds represent the 25th and 75th percentiles. Source data are provided as a Source Data file. **B** A Manhattan plot of the 99% SuSiE credible set colored by $r^2$ with the lead SNP. Black diamonds: SNPs in credible set 1. Black boxes: SNPs in credible set 2. Source data are provided as a Source Data file. **C** ATAC-seq peak regions are represented as boxes. Points above the plot area represent SNPs 99% credible set (red = rs9525462). NK cell = natural killer cell. **D** Roadmap chromatin ChIP-seq for hg19 chr13:41768158-41769095 (yellow). Samples: E044, E039, and E047. **E, F** Boxplot center is the mean and box bounds represent the 25th and 75th percentiles. **E** Knock-out $N = 20$, control $N = 11$. One-sided $t$ test $p$-value = 0.003037 (**). Source data are provided as a Source Data file. **F** Knock-out $N = 18$, control $N = 9$. One-sided $t$ test $p$-value = 2.093 $\times 10^{-5}$ (***). Source data are provided as a Source Data file.

results suggest that the role of the POP proteins also play a role in human telomerase regulation. KBTBD6 and KBTBD7 are members of an E3 ubiquitin ligase complex. CRISPR/Cas9 deletion of the high-likelihood causal region affected the expression of both genes, but overexpression of *KBTBD6* alone affected telomere length. Our results suggest that increased expression of the KBTBD6-KBTBD7-Cul3 complex or altered complex stoichiometry affects telomere length.

In addition to the *KBTBD6/KBTBD7* signal, we observed association signals that we attribute to *RFWD3*, another E3 ubiquitin ligase, *PSMB4*, a component of the core proteasome, and *UBE2D2*, an E2 ubiquitin-conjugating enzyme. ATM and ATR are kinases that contribute to the DNA damage response and telomere length regulation, though phosphorylation targets with strong effects on telomere length regulation have remained elusive. Prior proteome analysis demonstrated that ATM/ATR regulate the ubiquitin-proteasome pathway in response to DNA damage and validated RFWD3 as an ATM/ATR substrate[61]. Our results underscore the importance of ubiquitination in telomere length regulation; future work examining whether ATM/ATR substrates regulating the ubiquitination-proteasome pathway affect telomere length may identify ATM/ATR substrates with important

roles in telomere length regulation. Furthermore, identification of the ubiquitination targets by these E3 ubiquitin ligases may reveal telomere length regulation mechanisms. Together, this work demonstrates the potential contribution of telomere length GWAS to understanding mechanisms underlying telomere length regulation. Future work extending the findings reported here and validating additional loci will increase our understanding of both the genetics and molecular mechanisms underlying telomere length regulation.

## Methods

### Studies and telomere length estimation

We incorporated four telomere-length GWAS with non-overlapping cohorts. Delgado et al. had 5075 samples from Bangladeshi individuals and telomere length was estimated using qPCR or Luminex-based assay. Dorajoo et al. had 23,096 samples from Singaporean Chinese individuals and telomere length was estimated using qPCR. Li et al. (2020) had 78,592 samples from European individuals and telomere length was estimated using qPCR. Taub et al. had 51,654 individuals of European ancestry, 5683 individuals of Asian ancestry, 29,260 individuals of African ancestry, and 18,019 individuals of Hispanic/Latino ethnicity. In that study telomere length was estimated bioinformatically from whole genome sequencing data using TelSeq[62].

### Meta-analysis

One concern with a meta-analysis approach was whether it is reasonable to compare summary statistics from GWAS where telomere length was estimated using different methods. Previous work determined that each method produces telomere length estimates that are highly correlated with Southern blot analysis[24,63,64] and in each study telomere length estimates were standardized prior to running the GWAS. We used GWAMA[65] to conduct a random effect meta-analysis that represents a total of 211,379 individuals. GWAMA automatically calculates the Cochran's $q$ statistic and $I^2$ statistic for each SNP as estimates for heterogenity. We report these statistics for our lead SNPs in Supplementary Data 2 and they are available for all analyzed SNPs in our summary statistics file (see Data Availability). Taub et al. stratified individuals from the Trans-Omics for Precision Medicine (TOPMed) program cohorts by ancestry group where individuals were broadly categorized as European, African, Asian, or Hispanic/Latino using HARE and we maintain language used from that study here for clarity. That study also defined an "Other" group which was not included in our analysis. We provide a list of TOPMed cohorts whose data are represented in the meta-analysis and the broad ancestral groups individuals were categorized as (Supplementary Data 1). A detailed enumeration of individuals over ancestry by TOPMed cohort was previously published in Supplementary Table 1 of Taub et al. SNP positions were converted to hg38 using LiftOver prior to meta-analysis. The Delgado et al. summary statistics were harmonized to the forward strand and palindromic SNPs were removed from this dataset. Loci were considered novel if there were no other reported sentinels within 1 megabase of the lead SNP in the signal.

Lead SNPs were identified by minimum $p$-value within a 2 megabase window. We examined all loci with at least one variant that was genome-wide significant ($p$-value $< 5 \times 10^{-8}$) and had a minor allele frequency > 0.0001. This excluded loci where the lead SNPs were rs903494390, rs976923370, rs990671169, rs982808930, rs992178597, rs961617801, and rs1324702094. The signal led by rs3131064 is near the *HLA* locus and due to the extensive linkage disequilibrium in this region, we expanded the width of this signal to 4.2 megabases.

We considered a signal novel if the lead SNP was not within 1 megabase of a previously reported lead SNP in a telomere length GWAS[6-24]. One special circumstance arose for the signal led by rs3131064 which we report as a distinct signal from the signal led by rs1150748. However, SNPs in the signal led by rs3131064 were genome-wide significant ($p$-value $< 5 \times 10^{-8}$) in previous telomere length GWAS[23]

and therefore we do not consider this a novel signal in our meta-analysis. A second special circumstance arose for the signal led by rs12241155 which was genome-wide significant ($p < 5 \times 10^{-9}$) in the TOPMed pooled analysis[24] but was not reported as a signal because it was not conditionally independent of the signal led by rs9420907 (data not shown in that manuscript). We report it as a signal here, but do not consider it a novel signal in our analysis.

### Replication analysis

To determine whether novel association signals may be supported by other telomere length GWAS findings, we examined the highest powered telomere length GWAS not included in our dataset[23]. Of our five novel signals, four of the lead SNPs were evaluated in the replication dataset. The unexamined lead SNP, rs958919990, did not have any proxy SNPs that could be used as no SNPs in the region had $r^2 > 0.9$ with rs958919990. $r^2$ was calculated using a multi-ancestry group of all TOPMed individuals included in the meta-analysis. We considered a SNP replicated if it had an association $p$-value $< 0.05/5 = 0.01$ in the replication dataset.

### Colocalization analysis

All colocalization analysis was conducted using the coloc package[30] using the coloc.abf() command with the prior probability that the SNP is shared between the two traits (p12) set to 1e−6 and that there were at least 1000 shared variants between the two datasets. For GTEx_v8[31] colocalization we evaluated all genes for which the lead SNP was a significant QTL in any of the 49 GTEx_v8 tissues (FDR < 0.05) (Supplementary Data 4–5). For colocalization with eQTLGen cis-eQTLs (version available 2019-12-11)[33] (Supplementary Data 6) and DICE cis-eQTLs (version available 2019-06-07)[34] (Supplementary Data 7) we evaluated all genes within a 2 megabase window centered on the lead SNP and the meta-analysis summary statistics were lifted down to hg19 using LiftOver to compare SNPs based on chromosome and position. The X-chromsome signals could not be evaluated for colocalization with eQTLGen data as that dataset is limited to autosomes. Colocalization was conducted using minor allele frequency, p-value, and the number of samples for eQTLGen. Minor allele frequency was estimated from TOPMed pooled across ancestries. For all other colocalization analyses effect size estimates and their standard errors were used. We report the posterior probability that there are two signals but they do not share a causal signal (PPH3) and the posterior probability that there are two signals and they do share a causal signal (PPH4) within the text, figures, and figure legends. Posterior probabilities for the cases that there is no signal in one or either of the datasets (PPH0, PPH1, and PPH2) are reported in the appropriate Supplementary Data (4-7). We considered cases where PPH4 > 0.7 to be colocalized except for colocalization analysis with DICE cis-eQTLs where we reduced this threshold to PPH4 > 0.5 to account for the reduced power in the dataset. For Manhattan plots colored by linkage disequilibrium, $r^2$ was calculated using a multi-ancestry group of all TOPMed individuals included in the meta-analysis.

### Visualizing sQTLs

RNA alignment information for each individual was extracted using SAMtools (version 1.16) in the GTEx_v8 cultured fibroblast samples on AnVIL. We extracted genotype information from GTEx_v8 for the corresponding individuals and plotted the average alignment depth at each base position (hg38) stratified by genotype using Matplotlib. Visualization of LeafCutter[66] splicing clusters was produced using LeafCutter exon-exon junction quantifications generated by GTEx_v8[31].

### TWAS

TWAS was conducted using FUSION[37] using a pre-trained weight model from the Young Finns Cohort[36] which was trained on whole blood samples ($N = 1264$) available from the Gusev lab (http://gusevlab.org/

projects/fusion/). The X-chromosome was excluded from this analysis. The pre-trained model evaluated 4700 genes and a significance threshold of $0.05/4,700 = 1.06 \times 10^{-5}$ was used.

## Variant fine-mapping

Due to the multi-ancestry nature of our meta-analysis we used individual-level data from TOPMed individuals spanning all four ancestries represented in our meta-analysis (European, Asian, African, and Hispanic/Latino) as our linkage disequilibrium reference. Despite the fact that TOPMed individuals represent the largest group in the meta-analysis, the mismatch between the linkage disequilibrium reference and meta-analysis summary statistics was problematic for SuSiE (susieR_0.12.16)[40]. Therefore, we used summary statistics from the pooled TOPMed GWAS[24] to estimate credible sets for all meta-analysis signals since this was an exact match (Supplementary Data 8) and generated a genotype correlation matrix using a random subset, preserving the proportion of ancestries, of 15,000 TOPMed individuals to manage SNP density. We did not use a minor allele frequency threshold for SNP inclusion. At two loci the signal was over 1 megabase wide and calculating the genetic correlation matrix exceeded the ability of computational resources on the premises. At 16 loci there was not sufficient signal in the TOPMed GWAS to predict a credible set. CAVIAR[58] requires specification of the assumed number of causal signals whereas SuSiE jointly models the likelihood of varying numbers of causal signals and converges on the highest likelihood case. Due to this assumption and the computational burden of running CAVIAR, we only ran CAVIAR on the *POP5* and *KBTBD6*/*KBTBD7* loci.

For the signal led by rs1411041, which we attributed to *KBTBD6* and targeted for CRISPR/Cas9 editing, we further fine-mapped the locus by intersecting the credible set SNPs with ATAC-seq peaks and with ChIP-seq data from Roadmap Epigenomics. ATAC-seq data were downloaded from ENCODE (identifiers: ENCFF058UYY, ENCFF333TAT, ENCFF421XIL, ENCFF470YYO, ENCFF558BLC, ENCFF748UZH, ENCFF751CLW, ENCFF788BUI, and ENCFF867TMP) or from ATACdb (Sample_1195, Sample_1194, Sample_1175, Sample_1171, Sample_1020, Sample_1021, Sample_1209, and Sample_1208). BEDTools was used to identify intersecting regions. Roadmap Epigenomic ChIP-seq data was visualized using the WashU Epigenome browser.

## GO enrichment analysis

All gene ontology (GO) enrichment analysis was conducted using PANTHER[44] overrepresentation test with the GO Ontology database (released on 2022-07-01) with the all *Homo sapiens* gene set list as the reference list. PANTHER GO biological process complete terms were tested for enrichment using a Fisher's exact test with false discovery rate correction. Proximal genes were assigned as the gene with minimal distance to the gene body in the UCSC genome browser.

## Transcription factor binding site analyses

To assess the enrichment of 95% credible set SNPs with transcription factor and chromatin regulator DNA binding sites, we downloaded the ENCODE regulation track transcription factor binding site cluster ChIP-seq index file to report data for 330 DNA binding proteins spanning 129 cell types. The intersection of variants with transcription factor binding sites was performed by BEDTools v2.29.2. We computed the enrichment of 95% credible set SNPs in transcription factor binding sites using a GREGOR Perl based pipeline[67]. Briefly, this pipeline sums independent binomial random variables for the number of index SNPs falling in a single feature and calculates the enrichment *p*-value using a saddlepoint approximation method. The SNPs are considered to have a positional overlap if the input SNP, or variants in high linkage disequilibrium ($r^2$) with the input SNP ($r^2 > 0.7$, linkage disequilibrium window size = 1 megabase), fall within the regulatory features or overlap by $\geq 1$ base pair. The pairwise linkage disequilibrium was computed using the 1000 Genomes European reference panel.

Transcription factor binding site fold enrichment is measured as the fraction of index SNPs (or SNPs in linkage disequilibrium) overlapping the feature (as observed) over the mean number of overlaps with the control set of SNPs (as expected). Control SNPs are matched based on the number of variants in linkage disequilibrium, minor allele frequency, and distance to the nearest gene of the index SNPs. We also performed the enrichment analysis of 95% credible set SNPs with 1210 DNA-associated factors spanning across 737 cell-tissue types using the peak bed files downloaded from the ReMap 2022 database using the same pipeline. In addition, we performed both the ENCODE and ReMap enrichment analyses using only the lead SNP at each signal (Supplementary Fig. 4B, C). In addition to the enrichment analysis, we identified transcription factor binding sites overlapping the lead SNP for each meta-analysis association signal by searching the rsID on the UCSC genome browser and identified overlapping binding sites using the JASPAR 2022 track with default settings. We identified transcription factors with known roles in telomere length regulation by searching PubMed. Publication references supporting known roles for these transcription factors are indicated in Supplementary Data 10.

## Telomere length GWAS with an age × genotype interaction term

We repeated the pooled analysis from Taub et al. (2022) using all 109,122 TOPMed individuals with telomere length estimates. We ran the GWAS including an interaction term for genotype and age in addition to cohort, sequencing center, sex, age at sample collection, and 11 genotype PCs as covariates on Analysis Commons.

## Age-stratified GWAS

We divided the 109,122 TOPMed individuals with telomere length estimates into three age bins: ages 0–43 years old, ages 43.1–61 years old, and 61.1–98 years old. We ran the GWAS including cohort, sequencing center, sex, age at sample collection, and 11 genotype PCs as covariates on Analysis Commons. TOPMed cohorts included in this analysis are indicated in Supplementary Data 1. There were 36,980 individuals in the [0,43] group, 37,470 individuals in the (43,61] group, and 34,671 individuals in the (61,98] group. Any peak that cleared genome-wide significance ($p < 5 \times 10^{-8}$) in at least one age group was considered. We then required that the lead SNP in the signal was evaluated in all three age groups. To ensure a reasonable comparison between groups, we required that the minor allele count for the SNP was at least half of the maximum group minor allele count in each group. Then we identified loci where the effect size estimate confidence interval was non-overlapping in at least one age group. Finally, we examined loci that had a genotype × age interaction *p*-value < 5 × $10^{-5}$ and had a meta-analysis association *p*-value < 5 × $10^{-8}$.

## Enrichment of meta-analysis signals in chromatin states

We estimated the enrichment of lead meta-analysis signal SNPs across each state of the 25-state chromatin state model from Roadmap Epigenomics[29] across all 127 Roadmap Epigenomics samples (Supplementary Data 14). Similarly, Roadmap Epigenomics consolidated narrowPeak files for H3K4me1 and H3K27ac from 98 and 127 samples, respectively (Supplementary Data 14), were used to compute the enrichment of lead SNPs in ChIP-seq peak regions for these histone modifications. Control SNPs were randomly selected from the genome and matched for the number of linkage disequilibrium proxy SNPs, the minor allele frequency, and the distance to the nearest gene. The same GREGOR Perl script pipeline[67] used to evaluate transcription factor binding site enrichment (above) was used for these analyses. This script sums binomial random variables corresponding to the count of index SNPs located within any given states/features, followed by the computation of enrichment p-values via saddlepoint approximation.

To identify the cell type group with the strongest enrichment for each chromatin state we used Fisher's method to calculate a combined chi-squared statistic for the samples in each cell type group. We then

identified the group with the strongest enrichment for each chromatin state as the group with the smallest p-value. Briefly, active chromatin states may be considered states 1-19. For a full description of the chromatin states see the section on the 25 state model https://egg2.wustl.edu/roadmap/web_portal/imputed.html#chr_imp[29].

## Partitioned heritability across cell types (S-LDSC)

We limited our analysis to European individuals because the accuracy of this method depends upon an accurate match with the linkage disequilibrium reference panel. Therefore, we meta-analyzed the European individuals from two studies included in our meta-analysis[22,24] using GWAMA as described above and ran stratified linkage disequilibrium score regression (S-LDSC, 1.0.1) using the cell-type specific analyses pipeline. We directly used the 1000 Genomes European baseline files, multi-tissue gene expression counts, and multi-tissue chromatin marker data generated as part of the S-LDSC pipeline[28].

## Molecular cloning

Gibson assembly primers were designed using Snapgene software (GSL Biotech) and sequencing primers were identified using the GenScript sequencing primer tool. All primers were synthesized by IDT. Primer sequence and a brief description of their use are provided in Supplementary Data 15. Polymerase chain reaction products were amplified using Phusion HS II DNA polymerase (F549; Thermo Fisher). Gibson Assembly was conducted using Gibson Assembly Master Mix (E2611; NEB) according to the recommended protocol. Plasmids were transformed into NEB5α cells (C2987; NEB), prepared using the QIAprep Miniprep Kit (27104; Qiagen) or the Qiagen Plasmid Midiprep Kit (12143; Qiagen), and sequence verified using the Sanger method at the Johns Hopkins School of Medicine Synthesis & Sequencing Facility.

## Identifying candidate genes for overexpression experiments

We identified genes with eQTLs in any GTEx tissue that colocalized with our meta-analysis signals at a reduced threshold of PPH4 > 0.5. Elsewhere in this manuscript we used a threshold of PPH4 > 0.7, which we would consider to be strong colocalization. However, since we planned to experimentally validate the genes, we lowered the threshold to expand our candidate gene list. Next we required that tested allele of the lead SNP at the meta-analysis signal also have a significant effect (FDR < 0.05 in GTEx) on the expression of the candidate gene and be associated with increasing gene expression as the allele copy number increased in GTEx (the eSNP estimated effect size must be positive). In cases where the meta-analysis signal colocalized with the eQTL of a gene in multiple GTEx tissues, we examined the estimated effect size in the tissue where colocalization was strongest (greatest PPH4). Finally, to implement our overexpression experiment we would need to clone the gene into a plasmid, which limited the genes to those with a single protein isoform or with a gene length of less than 15 kilobases. We manually queried each gene on the NCBI RefSeq database and identified the number of known protein isoforms and obtained the gene length from the UCSC genome browser. Some genes, such as CBX1 and POP5, had multiple transcriptional isoforms that diverged in the untranslated regions. Because we planned to only overexpress the coding sequence, we counted these cases as a single protein isoform. Of the genes that met these criteria, we conducted a literature search on PubMed to determine whether the candidate genes had known roles in telomere biology or related processes and chose five. A detailed walkthrough of this filtering process is described in Supplementary Note 2.

## Overexpression constructs

All cDNA sequences were ordered through GenScript (OHu26641, OHu13170, OHu31184, OHu26125, OHu108607) with the coding sequence subcloned into a pcDNA3.1/C-DYK vector. We added the FLAG tag to the N- or C-terminus in accordance with precedent in the

literature: CBX1 C-terminus[68], PSMB4 C-terminus[69], POP5 N-terminus[70], OBFC1 N-terminus[71], and KBTBD6 N-terminus[72]. We used Gibson Assembly to add a 3x FLAG tag to the appropriate end and insert the tagged coding sequence into a pcDNA5/FRT vector (Thermo Fisher). We note that we overexpressed the propeptide of PSMB4 (removing amino acids 2–45). Plasmid maps are available at Zenodo (doi: 10.5281/zenodo.10476137).

## Cell culture

HeLa-FLP cells were generated from HeLa cells using the FLP-in system and were cultured in 1x Dulbecco's modified Eagle's medium (11965118; Thermo Fisher). K562 cells were purchased from ATCC (CCL-243) and were cultured in 1x RPMI medium (11875119; Thermo). Cells were cultured in the indicated media supplemented with 10% heat-inactivated fetal bovine serum (16140071; Thermo Fisher) and 1% Penicillin-Streptomycin-Glutamine (10378016; Thermo Fisher).

## Overexpression experiments and passaging

For overexpression experiments, 100 ng of the indicated overexpression construct and 900 ng of the pOG44 flippase plasmid were co-transfected into HeLa-FLP cells by the use of the FLP-in system using Lipofectamine 3000 (L3000008; Invitrogen) with the recommended protocol and hygromycin resistant (550 µg/mL; 30-240-CR; Corning) cells were examined. The GFP overexpression plasmid (pAMP0605) was previously generated[73]. For each construct, we used one pool of HeLa-FLP cells to conduct multiple independent transfections, which we refer to as independent clones. Twice a week, cells were treated with 0.05% trypsin-EDTA (25300054; Invitrogen), washed in 1x PBS (10010049; LifeTech), and counted using a Luna II Automated Cell Counter (Logos Biosystems). The number of population doublings for each passage was estimated as the number of cells counted divided by the number of cells seeded for that passage.

## Telomere Southern blot analysis

For each time point, $(2–4) \times 10^6$ cells were collected, washed in 1x PBS (10010049; LifeTech), and pellets stored at −80 °C. Genomic DNA was isolated using the Promega Wizard gDNA kit (A1120; Promega) as directed. Genomic DNA was quantified using the broad range double-stranded DNA kit (Q32853; Thermo Fisher) for QuBit 3.0 (Thermo Fisher). Approximately 1 µg of genomic DNA was restricted with HinfI (R0155M; NEB) and RsaI (R0167L; NEB) and resolved by 0.8% Tris-acetate-EDTA (TAE) agarose gel electrophoresis. 10 ng of a 1kB Plus DNA ladder (N3200; NEB) was included on either side of the Southern as a size reference. Following denaturation (0.5 M NaOH, 1.5 M NaCl) and neutralization (1.5 M NaCl, 0.5 M Tris-HCL, pH 7.4), the DNA was transferred in 10x SSC (3 M NaCl, 0.35 M NaCitrate) to a Nylon membrane (RPN303B; GE Healthcare) by vacuum blotting (Boekel Scientific). The membrane was UV crosslinked (Stratagene), prehybridized in Church buffer (0.5 M Na2HPO4, pH7.2, 7% SDS, 1 mM EDTA, 1% BSA), and hybridized overnight at 65 °C using a radiolabelled telomere fragment and ladder, as previously described[74]. Briefly, a 100x human telomere repeat fragment is excised by EcoRI restriction digest from JHU821 (aka pBLRep4) (Supplemental Data 15) and used for random isotope labeling with αP32 dGTP or dCTP (Thermo Fisher). The membrane was washed twice with a high salt buffer (2x SSC, 0.1% SDS) and twice with a low salt buffer (0.5X SSC, 0.1% SDS) at 65 °C, exposed to a Storage Phosphor Screen (GE Healthcare), and scanned on a Storm 825 imager (GE Healthcare). The images were copied from ImageQuant TL (GE Life Sciences) to Adobe PhotoShop CS6, signal was adjusted across the image using the curves filter, and the image was saved as a.tif file. Minimum, maximum and median telomere length was estimated in ImageQuant TL using the original, unedited scan from the Phosphor Screen and accounted for differences in DNA migration across the gel by including the 1 kB Plus ladder on either side of the Southern blot.

## Western blot analysis

$2 \times 10^6$ cells were collected, washed in 1x PBS (10010049; LifeTech), resuspended in 1x sample buffer (1x NuPAGE loading buffer (NP0008; Thermo Fisher), 50 µM DTT) and stored at −80 °C. Samples were thawed on wet ice, lysed by sonication, and boiled at 65 °C for 10 min. Proteins were resolved using recommended parameters on 4–12% Bis−Tris NuPAGE pre-cast gels (NP0321BOX; Invitrogen) and Precision Plus Dual Color protein ladder (161-0374; BioRad) was run for comparison. Proteins were transferred to a PVDF membrane (170-4273; BioRad) using a Trans-Blot Turbo Transfer System (BioRad). The membrane was blocked in 5% milk-TBST (w/v powdered milk (170-6404; BioRad) resuspended in 1x Tris Buffered Saline, pH 7.4 (351-086-101CS; Quality Biological), 0.01% Tween-10 (P1379-100ML; Sigma) for 1 h at room temperature. Primary antibodies were diluted in blocking buffer and incubated at room temperature for 1 h with mild agitation (M2 FLAG 1:2,000 (F1804-5MG; Sigma), tubulin 1:5,000 (ab6046; Abcam)). Blots were washed in 1x TBST with mild agitation before incubation with horseradish peroxidase-conjugated secondary antibodies diluted in blocking buffer (α-mouse 1:10,000 (170-6516; BioRad), α-rabbit 1:10,000 (170-6515; BioRad)). Blots were washed in 1x TBST with mild agitation, incubated with Forte horseradish peroxidase substrate (WBLUF0100; Millipore) for 5 min with agitation, and imaged on an ImageQuant LAS 4000 mini biomolecular imager (GE Healthcare). Image files were copied from ImageQuant TL software to Adobe PhosShop CS6, the curves filter was applied across the image, and then saved as a.tif file. To reprobe a membrane with the loading control, the membrane was incubated with Restore Western Blot Stripping Buffer (21059; Thermo Fisher) for 30 min, washed in 1x TBST, and processed as described above.

## CRISPR editing constructs

We sequence verified the CRISPR target regions in our K562 cells and selected gRNA sequences with a high likelihood of on-target editing (and a low likelihood of off-target editing) using CRISPOR.org. We subcloned the guides into px458 as previously described[75]. To edit both the *POP5* and *KBTBD6/KBTBD7* regions we chose one guide to each side of the target region (Supplementary Fig. 9C, D). For guide sequence and genome coordinates (hg38), see Supplementary Data 15.

## CRISPR editing experiments

Low-passage K562 cells were cultured to a density of $3 \times 10^5$ cells/mL in media without antibiotics, but otherwise as described above, two days prior to nucleofection. Cells were electroporated using the SF Cell Line 4D-Nucleofector X Kit (V4XC-2012; Lonza) with 8 µg of each guide plasmid and the K562 cell line recommended protocol (FF-120). Cells were cultured in antibiotic-free media for 24 h to allow for GFP expression before being single-cell sorted in a 96-well plate at the Johns Hopkins Ross Flow Cytometry Core. Each sample had 1–10% GFP-positive cells. Plates were expanded clonally using media described above. After approximately two weeks cell concentration was estimated using the Luna II Automated Cell Counter (Logos Biosystems), $4 \times 10^4$ cells were collected, and genomic DNA was extracted using QuickExtract DNA Extraction Solution (QE09050; Epicentre) following the protocol recommended in the Alt-R genomic editing detection kit (1075931; IDT). Target editing regions were amplified (primers described in Supplementary Data 15, diagrams in Supplementary Fig. 9) and confirmed by Sanger sequencing. Sequencing reads were aligned in Snapgene (GSL Biotech) and we considered a clone to at least be heterozygous for editing if the alignment began on one side of the deletion, failed across the intended deletion, but resumed across the deletion. Because the *POP5* locus deletion was so extensive, we did two separate PCRs on each sample: one that would amplify if the deletion was present (RK236 + RK231) and one that would amplify if a wildtype allele was present (RK236 + RK234) (Supplementary Fig. 9C). All *POP5* edited clones were confirmed to be heterozygous.

## RNA extraction and qPCR

$2 \times 10^6$ cells were collected, washed in 1x PBS (10010049; LifeTech), and RNA was purified using a QIAshreddar column (79656; Qiagen) and RNeasy kit (74104; Qiagen) following the recommended protocols, including DNase digestion of RNA prior to RNA cleanup (79254; Qiagen). RNA concentration was estimated using a high sensitivity RNA kit (Q32852; Thermo Fisher) for QuBit 3.0 (Thermo Fisher). cDNA was generated with random hexamers using a SuperScript IV First Strand Synthesis kit (18091050; Thermo Fisher). qPCR primers were designed using the GenScript RT-PCR primer design tool and a standard reference plasmid was generated by amplifying genomic DNA from K562 cells with each primer pair followed by TA cloning the amplicon into a pCR2.1 vector (Supplementary Data 15) using a TA cloning kit (451641; Thermo Fisher). TA cloning was conducted using the recommended protocol and plasmids were transformed into TOP10 cells (C404003; Invitrogen). Each qPCR reaction included approximately 10 ng of cDNA, 1x iQ SYBER Green Super Mix (1708882; BioRad), and 0.25 µM of each primer; qPCR was conducted on a CFX96 real-time qPCR system (BioRad). *KBTBD6* and *KBTBD7* expression was measured in the *POP5*-edited clones as CRISPR/Cas9-edited controls and *POP5* expression was measured in the *KBTBD6/KBTBD7*-edited clones as CRISPR/Cas9-edited controls. Samples were analyzed in triplicate and instances where the Cq range was greater than 1 were excluded from further analysis. Standard plasmids were analyzed in duplicate on each plate at a range of 0.001–100 ng as a quality control measure and plates where the standards Cq had an $R^2 < 0.98$ were excluded from further analysis. Plates that passed this threshold were used to estimate the efficiency of the qPCR primers ($ACTB = 1.90$, $KBTBD6 = 1.98$, $KBTBD7 = 1.92$, and $POP5 = 1.80$). Because the range of efficiency between measured genes was greater than 10%, we analyzed our qPCR results with the Pfaffl method. A one-sided $t$-test was used to compare experimental to control samples.

## Reporting summary

Further information on research design is available in the Nature Portfolio Reporting Summary linked to this article.

## Data availability

All cell lines and plasmids are available upon request. Summary statistics, plasmid maps, unprocessed blot images, and analysis dependent files are available at Zenodo (https://doi.org/10.5281/zenodo.10476137)[76] and are freely available. TOPMed genomic data and telomere length estimates are available by study in the database of Genotypes and Phenotypes (dbGaP) (https://www.ncbi.nlm.nih.gov/gap/?term=TOPMed). GTEx_v8 eQTL, sQTL, and LeafCutter exon-exon junction quantifications are available for download through the GTEx portal (https://gtexportal.org/home/). eQTLGen cis-eQTL data are available for download (https://www.eqtlgen.org/). In this manuscript we used the version available 2019-12-11. DICE cis-eQTL data are available for download (https://dice-database.org/landing). In this manuscript we used the version available 2019-06-07. Roadmap Epigenomics data can be visualized and downloaded here: https://egg2.wustl.edu/roadmap/web_portal/. ATAC-seq downloaded from ENCODE can be found here: https://www.encodeproject.org/. ATACdb data can be downloaded here: https://bio.tools/atacdb. ENCODE transcription factor ChIP-seq track data can be downloaded here (340 factors in 129 cell types from ENCODE 3): https://genome.ucsc.edu/cgi-bin/hgTrackUi?hgsid=1997834034_KyrOSyi5TZL4ybD9G2z2TOU6TCkR&c=chr7&g=encTfChipPk. ReMap 2022 data can be downloaded here: https://remap2022.univ-amu.fr/about_hsap_page. JASPAR 2022 transcription factor binding site data can be downloaded here: https://genome.ucsc.edu/cgi-bin/hgTrackUi?hgsid=1997834034_KyrOSyi5TZL4ybD9G2z2TOU6TCkR&db=hg38&c=chr7&g=jaspar. Source data are provided with this paper.

## Code availability

Code is available here: https://github.com/RKeener/telomere_length_metaanalysis, https://github.com/BennyStrobes/leafcutter_sqtl_viz, https://github.com/bulik/ldsc, https://github.com/stephenslab/susieR. Code and dependent files to reproduce figures in this manuscript are available at Zenodo (https://doi.org/10.5281/zenodo.10476185)[77]. Any additional information required to reanalyze the data reported here is available upon request.

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

## Acknowledgements

The authors thank Chen Li, Claudia Langenberg, Veryan Codd, Dayana Delgado, Brandon Pierce, and Rajkumar Dorajoo, in addition to all the individuals who were sampled, for providing summary statistics from their telomere length GWAS that were included in this meta-analysis. The whole-genome sequencing for the Trans-Omics in Precision Medicine (TOPMed) program was supported by the National Heart, Lung, and Blood Institute (NHLBI). Specific funding sources for each study and genomic center are given in the Supplementary Acknowledgements. px458 was a gift from Andrew Holland's lab. We acknowledge the ENCODE Consortium and the following ENCODE production laboratories: Michael Snyder and J. Michael Cherry. We would also like to thank Margaret Strong, Emily DeBoy, the JHU Synthesis & Sequencing Facility, and the JHU Ross Flow Cytometry Core for their technical assistance, Andrew Holland and Emmanouil Tampakakis for helpful discussion about CRISPR/Cas9 editing experiments, and the Mathias, Greider, and Battle labs for helpful discussion throughout the course of this work. Rebecca Keener was supported in part by NIH/NIGMS grant 5K12GM123914. This work was carried out at the Advanced Research Computing at Hopkins (ARCH) core facility (rockfish.jhu.edu), which is supported by the National Science Foundation (NSF) grant number OAC1920103. Matthew P. Conomos was supported by R01HL-120393; contract HHSN268201800001I; U01 HL137162. Brian E. Cade was supported by R01HL153805. Barry I. Freedman was supported by R01 DK071891. Lifang Hou was supported by R01AG081244, R01AG069120. Marilyn J. Telen was supported by NHLBI (R01HL68959, R01HL87681 and R01HL079915) for the collection of the samples. Allison E. Ashley-Koch was supported by NHLBI (R01HL68959, R01HL87681 and R01HL079915) for the collection of the samples. Joshua C. Bis was supported by R01HL105756. Zhanghua Chen was supported by P30ES007048, 5P01ES011627, ES021801, ES023262, P01ES009581, P01ES011627, P01ES022845, R01 ES016535, R03ES014046, P50 CA180905,

R01HL061768, R01HL076647, R01HL087680, RC2HL101651, RD83544101, R826708, RD831861, R831845, R00ES027870, and the Hastings Foundation. Dr. Patrick T. Ellinor was supported by grants from the National Institutes of Health (1R01HL092577, 1R01HL157635, 5R01HL139731), from the American Heart Association Strategically Focused Research Networks (18SFRN34110082), and from the European Union (MAESTRIA 965286). Myriam Fornage was supported by U01AG052409, U01AG058589. Bruce D. Gelb was supported by U01HL153009. Frank D. Gilliland was supported by P30ES007048, 5P01ES011627, ES021801, ES023262, P01ES009581, P01ES011627, P01ES022845, R01 ES016535, R03ES014046, P50 CA180905, R01HL061768, R01HL076647, R01HL087680, RC2HL101651, RD83544101, R826708, RD831861, R831845, R00ES027870, and the Hastings Foundation. Talat Islam was supported by P30ES007048, 5P01ES011627, ES021801, ES023262, P01ES009581, P01ES011627, P01ES022845, R01 ES016535, R03ES014046, P50 CA180905, R01HL061768, R01HL076647, R01HL087680, RC2HL101651, RD83544101, R826708, RD831861, R831845, R00ES027870, and the Hastings Foundation. Rajesh Kumar was supported by UM1AI160040, U01AI160018-01, UG1HL139125, R01AI153239. Ruth J.F. Looks was supported by R01DK107786; R01HG010297. Nicholette D. Palmer was supported by R01 AG058921. Susan Redline was supported by R35HL135818. David Schwartz was supported by P01-HL162607, R01-HL158668, R01-HL149836, IO1BX005295, UG3/UH3-HL151865, and X01-HL134585. The Johns Hopkins Genetic Study of Atherosclerosis Risk (GeneSTAR) was supported by grants from the National Institutes of Health through the National Heart, Lung, and Blood Institute (U01HL72518, HL087698, HL112064) and by a grant from the National Center for Research Resources (M01-RR000052) to the Johns Hopkins General Clinical Research Center. Dr. Rasika Mathias receives support as the Sarah Miller Coulson Scholar in the Johns Hopkins Center for Innovative Medicine. Carol W. Greider was supported by NIH R35 CA209974. Alexis Battle was supported by NIH/NIGMS R35GM139580.

## Author contributions

R.Keener, C.W.G., R.Mathias, and A.B. conceived of and led the study. R.Keener, S.B.C., C.J.C., M.Taub, J.S.W., L.R.Y., L.M.R., A.P.R., C.W.G., R.Mathias, and A.B. drafted the manuscript. R.Keener, S.B.C., C.J.C, M.Taub, M.P.C., J.S.W., B.N., B.J.S., M.A., C.W.G., R.Mathias, and A.B. contributed substantive analytical guidance. R.Keener, S.B.C., C.J.C, M.Taub, M.P.C., J.S.W., B.N., B.J.S., M.A., C.W.G, R.Mathias, and A.B. performed and led the analysis. R.Keener, M.Taub, M.P.C., J.S.W., S.A., P.L.A., L.Barwick, L.Becker, J.Blangero, E.Bleecker, J.Brody, B.Cade, J.C.C., Y.C., L.A.C., B.Custer, B.I.F., M.T.G., S.R.H., L.H., M.R.I., C.R.I., J.M.J., E.E.K., C.K., R.L.Minster, T.N., S.V., S.N., N.Pankratz, P.A.P., J.I.R., K.D.T., M.Telen, B.W., L.R.Y., I.V.Y., C.A., D.K.A., A.E.A.K., K.C.B., J.Bis, T.W.B., E.Boerwinkle, E.Burchard, A.P.C., Z.C., Y.I.C., D.D., M.dA., P.T.E., M.F., B.D.G., F.D.G., J.H., T.I., S.Kaab, S.L.R.Kardia, S.Kelly, B.A.K., R.Kumar, R.J.F.L., F.D.M., S.T.M., D.A.M., B.D.M., C.G.M., K.E.N., N.D.Palmer, J.M.P., B.A.R., S.Redline, S.Rich, D.R., I.R., D.S., F.S., M.B.Shoemaker, E.K.S., M.F.Sinner, N.L.S., A.V.S., H.K.T., R.S.V., S.T.W., L.K.W., Y.Z., E.Z., L.M.R., A.P.R., M.A., R.Mathias, and A.B. were involved in the guidance, collection, and analysis of one or more of the studies that contributed data to this article. All authors read and approved the final draft. Meta-analysis, replication analysis, colocalization analysis, TWAS, fine-mapping, GO enrichment analysis, age-stratified GWAS, and S-LDSC were conducted by R. Keener and overseen by A.B. Age by genotype interaction was conducted by R. Keener with the guidance of M.Taub and M.P.C. and overseen by R.Mathias and A.B. Transcription factor binding analysis and the enrichment of meta-analysis lead SNPs over chromatin states was conducted by S.B.C with the guidance of R. Keener and overseen by A.B. Identification of candidate genes for overexpression analysis was conducted by R. Keener and overseen by C.W.G. and A.B. Design of experimental validation experiments was conducted by R. Keener and overseen by C.W.G. and A.B. Molecular cloning, overexpression experiments, cell culture, western blot analysis, CRISPR editing, and qPCR were conducted by R. Keener and overseen by C.W.G. Telomere Southern blot analysis was conducted by R.Keener and C.J.C. and overseen by C.W.G.

## Competing interests

The authors declare the following competing interests: Juan C. Celedon received inhaled steroids from Merck for an NIH_funded study, unrelated to this work. Ivana V. Yang is a consultant for Eleven P15, a company focused on the early diagnosis and treatment of lung fibrosis. Dr. Patrick T. Ellinor receives sponsored research support from Bayer AG, IBM Research, Bristol Myers Squibb and Pfizer; he has also served on advisory boards or consulted for Bayer AG, MyoKardia and Novartis. Dr. David Schwartz is a founder and chief scientific officer of Eleven P15, a company focused on the early diagnosis and treatment of lung fibrosis. Laura M. Raffield is a consultant for the TOPMed Admistrative Coordinating Center (through Westat). Alexis Battle is a shareholder in Alphabet, Inc.; consultant for Third Rock Ventures, LLC; and founder of CellCipher, Inc. The views expressed in this manuscript are those of the authors and do not necessarily represent the views of the National Heart, Lung, and Blood Institute; the National Institutes of Health; or the U.S. Department of Health and Human Services. The remaining authors declare no competing interests.

## Additional information

Rebecca Keener [1], Surya B. Chhetri[1], Carla J. Connelly[2], Margaret A. Taub[3], Matthew P. Conomos [4], Joshua Weinstock[1], Bohan Ni[5], Benjamin Strober [6], Stella Aslibekyan[7], Paul L. Auer [8], Lucas Barwick[9], Lewis C. Becker[10], John Blangero[11], Eugene R. Bleecker[12,13], Jennifer A. Brody [14], Brian E. Cade [15,16], Juan C. Celedon [17], Yi-Cheng Chang[18,19,20], L. Adrienne Cupples [21,22], Brian Custer [23,24], Barry I. Freedman [25], Mark T. Gladwin [26], Susan R. Heckbert [27], Lifang Hou[28], Marguerite R. Irvin[29], Carmen R. Isasi[30], Jill M. Johnsen [31], Eimear E. Kenny[32,33], Charles Kooperberg [34], Ryan L. Minster[35], Take Naseri[36,37], Satupa'itea Viali[38,39,40], Sergei Nekhai[41], Nathan Pankratz[42], Patricia A. Peyser [43], Kent D. Taylor[44], Marilyn J. Telen [45], Baojun Wu[46], Lisa R. Yanek[47], Ivana V. Yang[48], Christine Albert[49,50], Donna K. Arnett[51], Allison E. Ashley-Koch [45], Kathleen C. Barnes[52], Joshua C. Bis[14], Thomas W. Blackwell[53,54], Eric Boerwinkle[55], Esteban G. Burchard [56,57], April P. Carson [58], Zhanghua Chen[59], Yii-Der Ida Chen [44], Dawood Darbar [60], Mariza de Andrade[61], Patrick T. Ellinor [62], Myriam Fornage [63], Bruce D. Gelb [64], Frank D. Gilliland [59], Jiang He[65], Talat Islam[59], Stefan Kaab [66], Sharon L. R. Kardia[67], Shannon Kelly[23,68], Barbara A. Konkle[69], Rajesh Kumar[70,71], Ruth J. F. Loos [32], Fernando D. Martinez[72], Stephen T. McGarvey [73], Deborah A. Meyers[12,13], Braxton D. Mitchell [74], Courtney G. Montgomery [75], Kari E. North [76], Nicholette D. Palmer [77], Juan M. Peralta [11], Benjamin A. Raby[78,79], Susan Redline [15,16], Stephen S. Rich [80], Dan Roden [81], Jerome I. Rotter [44], Ingo Ruczinski[3], David Schwartz[82], Frank Sciurba[83], M. Benjamin Shoemaker[84], Edwin K. Silverman[15], Moritz F. Sinner [66], Nicholas L. Smith[65], Albert V. Smith [85], Hemant K. Tiwari[86], Ramachandran S. Vasan [87], Scott T. Weiss[15,49], L. Keoki Williams[46], Yingze Zhang [88], Elad Ziv [56], Laura M. Raffield [89], Alexander P. Reiner [34], NHLBI Trans-Omics for Precision Medicine (TOPMed) Consortium*, TOPMed Hematology and Hemostasis Working Group*, TOPMed Structural Variation Working Group*, Marios Arvanitis[90], Carol W. Greider [91,92], Rasika A. Mathias [47,96] ✉ & Alexis Battle [1,5,93,94,95,96] ✉

[1]Department of Biomedical Engineering, Johns Hopkins University, Baltimore, MD, USA. [2]Department of Molecular Biology and Genetics, Johns Hopkins University, Baltimore, MD, USA. [3]Department of Biostatistics, Johns Hopkins Bloomberg School of Public Health, Baltimore, MD, USA. [4]Department of Biostatistics, School of Public Health, University of Washington, Seattle, WA, USA. [5]Department of Computer Science, Johns Hopkins University, Baltimore, MD, USA. [6]Department of Epidemiology, Harvard School of Public Health, Boston, MA, USA. [7]University of Alabama at Birmingham, Birmingham, AL, USA. [8]Division of Biostatistics, Institute for Health & Equity, and Cancer Center, Medical College of Wisconsin, Milwaukee, WI, USA. [9]LTRC Data Coordinating Center, The Emmes Company, LLC, Rockville, MD, USA. [10]GeneSTAR Research Program, Department of Medicine, Johns Hopkins School of Medicine, Baltimore, MD, USA. [11]Department of Human Genetics and South Texas Diabetes and Obesity Institute, University of Texas Rio Grande Valley School of Medicine, Brownsville, TX, USA. [12]Department of Medicine, Division of Genetics, Genomics and Precision Medicine, University of Arizona, Tucson, AZ, USA. [13]Division of Pharmacogenomics, University of Arizona, Tucson, AZ, USA. [14]Cardiovascular Health Research Unit, Department of Medicine, University of Washington, Seattle, WA, USA. [15]Channing Division of Network Medicine, Department of Medicine, Brigham and Women's Hospital, Boston, MA, USA. [16]Division of Sleep Medicine, Harvard Medical School, Boston, MA, USA. [17]Division of Pediatric Pulmonary Medicine, University of Pittsburgh, Pittsburgh, PA, USA. [18]Department of Internal Medicine, National Taiwan University, Taipei, Taiwan. [19]Graduate Institute of Medical Genomics and Proteomics, National Taiwan University, Taipei, Taiwan. [20]Institute of Biomedical Sciences, Academia Sinica, Taipei, Taiwan. [21]Department of Biostatistics, Boston University School of Public Health, Boston, MA, USA. [22]The National Heart, Lung, and Blood Institute, Boston University's Framingham Heart Study, Framingham, MA, USA. [23]Vitalant Research Institute, San Francisco, CA, USA. [24]Department of Laboratory Medicine, University of California San Francisco, San Francisco, CA, USA. [25]Internal Medicine - Nephrology, Wake Forest University School of Medicine, Winston-Salem, NC, USA. [26]School of Medicine, University of Maryland, Baltimore, MD, USA. [27]Department of Epidemiology, University of Washington, Seattle, WA, USA. [28]Department of Preventive Medicine, Feinberg School of Medicine, Northwestern University, Evanston, IL, USA. [29]Department of Epidemiology, University of Alabama Birmingham, Birmingham, AL, USA. [30]Department of Epidemiology and Population Health, Albert Einstein College of Medicine, Bronx, NY, USA. [31]Department of Medicine and Institute for Stem Cell & Regenerative Medicine, University of Washington, Seattle, WA, USA. [32]The Charles Bronfman Institute for Personalized Medicine, Icahn School of Medicine at Mount Sinai, New York, NY, USA. [33]Center for Genomic Health, Icahn School of Medicine at Mount Sinai, New York, NY, USA. [34]Division of Public Health Sciences, Fred Hutchinson Cancer Center, Seattle, WA, USA. [35]Department of Human Genetics, University of Pittsburgh Graduate School of Public Health, Pittsburgh, PA, USA. [36]Naseri & Associates Public Health Consultancy Firm and Family Health Clinic, Apia, Samoa. [37]International Health Institute, School of Public Health, Brown University, Providence, RI, USA. [38]Oceania University of Medicine, Apia, Samoa. [39]School of Medicine, National University of Samoa, Apia, Samoa. [40]Department of Chronic Disease Epidemiology, Yale University School of Public Health, New Haven, CT, USA. [41]Center for Sickle Cell Disease and Department of Medicine, College of Medicine, Howard University, Washington DC, USA. [42]Department of Laboratory Medicine and Pathology, University of Minnesota, Minneapolis, MN, USA. [43]Department of Epidemiology, School of Public Health, University of Michigan, Ann Arbor, MI, USA. [44]The Institute for Translational Genomics and Population Sciences, Department of Pediatrics, The Lundquist Institute for Biomedical Innovation at Harbor-UCLA Medical Center, Torrance, CA, USA. [45]Department of Medicine, Duke University Medical Center, Durham, NC, USA. [46]Center for Individualized and Genomic Medicine Research (CIGMA), Department of Internal Medicine, Henry Ford Health System, Detroit, MI, USA. [47]Department of Medicine, Johns Hopkins University School of Medicine, Baltimore, MD, USA. [48]Departments of Biomedical Informatics, Medicine, and Epidemiology, University of Colorado, Boulder, CO, USA. [49]Harvard Medical School, Boston, MA, USA. [50]Division of Cardiovascular, Brigham and Women's Hospital, Boston, MA, USA. [51]Department of Epidemiology and Biostatistics, University of South Carolina, Columbia, SC, USA. [52]Department of Medicine, University of Colorado Denver, Anschutz Medical Campus, Aurora, CO, USA. [53]Department of Biostatistics, University of Michigan School of Public Health, Ann Arbor, MI, USA. [54]Center for Statistical Genetics, University of Michigan School of Public Health, Ann Arbor, MI, USA. [55]Human Genetics Center, Department of Epidemiology, Human Genetics, and Environmental Sciences, School of Public Health, University of Texas Health Science Center at Houston, Houston, TX, USA. [56]Department of Medicine, University of California San Francisco, San Francisco, CA, USA. [57]Department of Bioengineering and Therapeutic Sciences, University of California San Francisco, San Francisco, CA, USA. [58]Department of Medicine, University of Mississippi Medical Center, Jackson, MI, USA. [59]Department of Population and Public Health Sciences, University of Southern California, Los Angeles, CA, USA. [60]Division of Cardiology, University of Illinois at Chicago, Chicago, IL, USA. [61]Division of Biomedical

Statistics and Informatics, Mayo Clinic, Rochester, MN, USA. [62]Cardiovascular Disease Initiative, The Broad Institute of MIT and Harvard, Cambridge, MA, USA. [63]Institute of Molecular Medicine, McGovern Medical School, the University of Texas Health Science Center at Houston, Houston, TX, USA. [64]Mindich Child Health and Development Institute and Departments of Pediatrics and Genetics and Genomic Sciences, Icahn School of Medicine, New York, NY, USA. [65]Department of Medicine, Tulane University School of Medicine, New Orleans, LA, USA. [66]Department of Cardiology, University Hospital, LMU Munich, Munich, Germany. [67]Department of Epidemiology, University of Michigan School of Public Health, Ann Arbor, MI, USA. [68]University of California San Francisco Benioff Children's Hospital, Oakland, CA, USA. [69]Department of Medicine, University of Washington, Seattle, WA, USA. [70]Northwestern University Feinberg School of Medicine, Chicago, IL, USA. [71]The Ann and Robert H. Lurie Children's Hospital of Chicago, Chicago, IL, USA. [72]Asthma & Airway Disease Research Center, University of Arizona, Tucson, AZ, USA. [73]Department of Epidemiology & International Health Institute, Brown University School of Public Health, Providence, RI, USA. [74]Department of Medicine, University of Maryland School of Medicine, Baltimore, MD, USA. [75]Genes and Human Disease, Oklahoma Medical Research Foundation, Oklahoma City, OK, USA. [76]Department of Epidemiology, University of North Carolina at Chapel Hill, Chapel Hill, NC, USA. [77]Department of Biochemistry, Wake Forest University School of Medicine, Winston-Salem, NC, USA. [78]Division of Pulmonary and Critical Care, Brigham and Women's Hospital, Boston, MA, USA. [79]Division of Pulmonary Medicine, Boston Children's Hospital, Boston, MA, USA. [80]Center for Public Health Genomics, Department of Public Health Sciences, University of Virginia, Charlottesville, VA, USA. [81]Department of Medicine, Vanderbilt University School of Medicine, Nashville, TN, USA. [82]Departments of Medicine and Immunology, University of Colorado, Boulder, CO, USA. [83]Division of Pulmonary, Allergy and Critical Care Medicine, University of Pittsburgh, Pittsburgh, PA, USA. [84]Departments of Medicine, Pharmacology, and Biomedical Informatics, Vanderbilt University Medical Center, Nashville, TN, USA. [85]Department of Biostatistics, University of Michigan, Ann Arbor, MI, USA. [86]Department of Biostatistics, University of Alabama Birmingham, Birmingham, AL, USA. [87]Department of Medicine, Boston University School of Medicine, Boston, MA, USA. [88]Division of Pulmonary Allergy and Critical Care Medicine, University of Pittsburgh, Pittsburgh, PA, USA. [89]Department of Genetics, University of North Carolina at Chapel Hill, Chapel Hill, NC, USA. [90]Department of Medicine, Division of Cardiology, Johns Hopkins University, Baltimore, MD, USA. [91]Department of Molecular Cell and Developmental Biology, University of California Santa Cruz, Santa Cruz, CA, USA. [92]University Professor Johns Hopkins University, Baltimore, MD, USA. [93]Department of Genetic Medicine, Johns Hopkins University, Baltimore, MD, USA. [94]Malone Center for Engineering in Healthcare, Johns Hopkins University, Baltimore, MD, USA. [95]Data Science and AI Institute, Johns Hopkins University, Baltimore, MD, USA. [96]These authors jointly supervised this work: Rasika A. Mathias, Alexis Battle. *Lists of authors and their affiliations appear at the end of the paper. ✉e-mail: rmathias@jhmi.edu; ajbattle@jhu.edu

## NHLBI Trans-Omics for Precision Medicine (TOPMed) Consortium

Rebecca Keener [1], Margaret A. Taub[3], Matthew P. Conomos [4], Joshua Weinstock [1], Stella Aslibekyan[7], Paul L. Auer [8], Lucas Barwick[9], Lewis C. Becker[10], John Blangero[11], Eugene R. Bleecker[12,13], Jennifer A. Brody [14], Brian E. Cade [15,16], Juan C. Celedon [17], Yi-Cheng Chang[18,19,20], L. Adrienne Cupples [21,22], Brian Custer [23,24], Barry I. Freedman [25], Mark T. Gladwin [26], Susan R. Heckbert [27], Lifang Hou[28], Marguerite R. Irvin[29], Carmen R. Isasi[30], Jill M. Johnsen [31], Eimear E. Kenny[32,33], Charles Kooperberg [34], Ryan L. Minster [35], Take Naseri[36,37], Satupa'itea Viali[38,39,40], Sergei Nekhai[41], Nathan Pankratz[42], Patricia A. Peyser [43], Kent D. Taylor[44], Marilyn J. Telen [45], Baojun Wu[46], Lisa R. Yanek [47], Ivana V. Yang[48], Christine Albert[49,50], Donna K. Arnett[51], Allison E. Ashley-Koch [45], Kathleen C. Barnes[52], Joshua C. Bis [14], Thomas W. Blackwell[53,54], Eric Boerwinkle[55], Esteban G. Burchard [56,57], April P. Carson [58], Zhanghua Chen[59], Yii-Der Ida Chen [44], Dawood Darbar [60], Mariza de Andrade[61], Patrick T. Ellinor [62], Myriam Fornage [63], Bruce D. Gelb [64], Frank D. Gilliland [59], Jiang He[65], Talat Islam[59], Stefan Kaab [66], Sharon L. R. Kardia[67], Shannon Kelly[23,68], Barbara A. Konkle[69], Rajesh Kumar[70,71], Ruth J. F. Loos [32], Fernando D. Martinez[72], Stephen T. McGarvey[73], Deborah A. Meyers[12,13], Braxton D. Mitchell [74], Courtney G. Montgomery[75], Kari E. North [76], Nicholette D. Palmer [77], Juan M. Peralta [11], Benjamin A. Raby[78,79], Susan Redline [15,16], Stephen S. Rich [80], Dan Roden [81], Jerome I. Rotter [44], Ingo Ruczinski[3], David Schwartz[82], Frank Sciurba[83], M. Benjamin Shoemaker[84], Edwin K. Silverman[15], Moritz F. Sinner [66], Nicholas L. Smith [65], Albert V. Smith [85], Hemant K. Tiwari[86], Ramachandran S. Vasan [87], Scott T. Weiss [15,49], L. Keoki Williams[46], Yingze Zhang [88], Elad Ziv [56], Laura M. Raffield [89], Alexander P. Reiner [34], Marios Arvanitis[90], Rasika A. Mathias [47,96] ✉ & Alexis Battle [1,5,93,94,95,96] ✉

## TOPMed Hematology and Hemostasis Working Group

Rebecca Keener [1], Margaret A. Taub[3], Matthew P. Conomos [4], Joshua Weinstock [1], Paul L. Auer [8], Lewis C. Becker[10], John Blangero[11], Jennifer A. Brody [14], L. Adrienne Cupples [21,22], Carmen R. Isasi[30], Jill M. Johnsen [31], Charles Kooperberg [34], Nathan Pankratz[42], Kent D. Taylor[44], Marilyn J. Telen [45], Baojun Wu[46], Lisa R. Yanek [47], Ivana V. Yang[48], Allison E. Ashley-Koch [45], Kathleen C. Barnes[52], Thomas W. Blackwell[53,54], Shannon Kelly[23,68], Braxton D. Mitchell [74], Stephen S. Rich [80], Nicholas L. Smith [65], Laura M. Raffield [89], Alexander P. Reiner [34] & Rasika A. Mathias [47,96] ✉

## TOPMed Structural Variation Working Group

Margaret A. Taub[3], Matthew P. Conomos [4], Joshua Weinstock [1], Paul L. Auer [8], Jill M. Johnsen [31], Charles Kooperberg [34], Ryan L. Minster [35], Nathan Pankratz [42], Patricia A. Peyser [43], Kathleen C. Barnes[52], Thomas W. Blackwell[53,54], Stephen S. Rich [80], Ingo Ruczinski[3], Albert V. Smith [85], Alexander P. Reiner [34] & Rasika A. Mathias [47,96] ✉

Lists of members and their affiliations appears in the Supplementary Information.

