## [Peer Review File · Nature Communications]

Validation of human telomere length multi-ancestry meta-analysis association signals identifies POP5 and KBTBD6 as novel human telomere length regulation genesREVIEWER COMMENTS

Reviewer #1 (Remarks to the Author):

In the study entitled "Validation of human telomere length trans-ancestry meta-analysis association signals identifies 2 POP5 and KBTBD6 as novel human telomere length regulation genes", Keener et al. identified 56 genome-wide significant loci ($P < 5e-8$) associated with leukocyte telomere length including 7 novel signals through trans-ancestry meta-analysis. Five genes (OBFC1, PSMB4, CBX1, KBTBD6, and POP5) were selected for validation experiments and two were demonstrated to be involved in telomere length regulation. As stated in the Abstract, the noteworthy result is that their work "validated these putative novel genes to confirm the contribution of GWAS loci to telomere length regulation" through statistical analyses such as trans-ancestry meta-analysis, heritability enrichment analyses and validation experiments.

However, the major concern with the study is that the link between statistical results in the first few sections and the subsequent validation experiments look weak and it is difficult to see how the statistical analyses directly lead to the validation experiments and choice of genes for validation. For example, the authors state on page 13 "Therefore, we identified candidate genes where the lead SNP was predicted to increase gene expression. Of those we chose five genes that had one known protein coding sequence isoform, had strong colocalization analysis results, and had some known biology: OBFC1, PSMB4, CBX1, KBTBD6, and POP5 (Methods)." However, one of the validated genes POP5 claimed by the authors is not even mentioned once in the colocalization sections which the majority of the paper is focussed on. Another validated gene KBTBD6 was mentioned without defining strong colocalization (page 10). Furthermore, none of the genes above were identified in the novel signals yielded from trans-ancestry meta-analysis (blue dots in Figure 1). They are mentioned in Supplementary Note 1 but not all the five genes have the characteristics (isoform, colocalisation, known biology), according to this supplementary note.

It would be important for the authors to clarify this as currently, it is quite unclear how the validation experiments directly lead on from the work earlier in the study. Given that the discussion section begins with how the GWAS meta-analysis and fine-mapping work is a useful platform for further experimental validation, it is prudent for the authors to have clear and reproducible selection criteria about how these elements came together to provide putative targets for validation.

Other comments:

1. The study identifies 2 - POP5 and KBTBD6 - as novel human telomere length regulation genes". However, POP5 and KBTBD6 are not among the 7 novel signals (Figure 1) by meta-GWAS. Is it appropriate for the authors to claim that they are (novelly) identified by trans-ancestry meta-analysis?
2. The authors outlined three criteria for identifying candidate genes for screening validation experiments in the Methods (page 22). More information is needed such as how a meta-analysis signal or the strongest colocalization was defined. This information may have been provided somewhere else in the manuscript/supplementary file. But it would be more clear if a summarising table (or descriptions of key details e.g. meta-analysis P values, colocalization definition) can be provided in the Results section accordingly, which would also serve as a benchmark for future studies adopting a similar strategy. Similarly, 'The strongest enrichment of several active chromatin states was observed in blood and T-cell samples' (page 13). Which threshold value was used for the 'strongest enrichment'? Threshold information ought to be provided wherever applicable.
3. The authors state the following criteria 'the lead variant at the meta-analysis signal was required to be associated with increased gene expression'. Does this mean that a lead variant should also be an eQTL? How can a variant be associated with a direction of gene expression? Shouldn't it be modelled as a specific allele (e.g., positive/negative-effect allele) of trait A (i.e., telomere length) that

promotes/reduces trait B (i.e., gene expression)? There are other similar descriptions elsewhere in the manuscript that should also be rephrased/clarified. Again, a threshold for such associations with gene expression ought to be provided.

4. I would suggest simplifying the Results especially for the statistical analyses – there appears to be extended description and discussions for colocalization analyses (4 sections) that are difficult to follow.

5. Is the finding that 'TCL1A 95% credible set SNPs are more strongly associated with telomere length in older individuals' relevant to the context? Otherwise it appears like a bit of a distraction.

6. Regarding the blood/T-cell enrichment (page 13), I would suggest that the authors rephrase the relevant content to improve readability. For instance, explanation of 'the chromHMM model is a predicted state'. Furthermore, what was the 'primary data for specific chromatin marks'?

7. In the S-LDSC analysis detecting heritability enrichment (page 13), shouldn't the cell type enrichment in blood cells be under expectation as the 'Telomere length was estimated from blood leukocytes'?

8. The authors found that "blood/immune cell category was the only category that was significantly enriched" (page 13). However, at the beginning of that section, the authors cited a published finding that "telomere length regulation is relevant in many different cell types to different extent" (page 12). Is the finding in the present study contradictory to the published one? How would the authors explain this?

9. In Figure 5B, does it suggest that the top enrichment signal (above the dash line) comes from the gene expression data in spleen cells, and others were not significant? What P-value does the dash line indicate? Are the labels on x-axis tissues or cell-types?

10. In page 13, "HeLa cells are not derived from blood or immune cells but are highly tractable for this screening stage of the validation experiments". This seems illogical and the choice of HeLa cells for validation seems to call into question the importance and relevance of the cell-type enrichment results?

11. Brief discussion of the potential clinical relevance of the study's findings should be included.

Reviewer #2 (Remarks to the Author):

This meta-analysis of several prior GWAS of TL reports 56 signals for leukocyte TL, 7 of which are novel. This paper also characterizes mechanisms using QTL data and conducts experiments to identify causal genes and regulatory elements for two TL association signals (POP5 and KBTBD6). This paper is an important contribution to the TL literature, but I have several concerns below that can hopefully be addressed.

Major Comments:

--"...overexpression of KBTBD6 or POP5 showed a clear telomere length increase over increased cell division, concordant with the expectation from our meta-analysis (Figure 6)." Looking at Figure 6 (B and D), it does not appear that this statement is accurate. I see decreases in TL at the third time-point for KBTBD6 clone 7 and for POP5 clones 5 and 6. I do see increases for the second time point (but not the third).

--when determining what signals are novel, what prior studies were included? Does this include the UK Biobank GWAS?

--Given UKB results are not included in this meta-analysis, is it possible to replicate the novel signals detected here using the UKB summary statistics?

--Replication would also help for the POP5 signal, which has a borderline P-value. Has POP5 been reported previously? Given the importance of POP5 for this paper, more is needed to convince readers the association is real (and not a false positive).

--In Fig 1, POP5 (and a few other regions) do not appear to cross the threshold, so readers will wonder why these are labeled as TL signals. Some clarification is needed.

--"Telomere length was estimated from blood leukocytes in all samples; it is possible that this boosted the strength of blood and immune cell enrichment in our analyses." This highly likely in my view. Also, this paragraph includes no mention of the fact that telomere maintenance by telomerase occurs primarily in stem and progenitor cells, not the differentiated cell/tissue types that make up most of GTEx, ENCODE, etc.. This seems very relevant to the 1st paragraph of the discussion.

--Discussion: "Together, these results suggest that the role of the POP proteins also play a role in human telomerase regulation" is telomerase active in the cell type(s) used for experimental validation?

--is it possible to attempt replication in an independent data source of the SNP-by-Age interaction for TCL1A? And does the eQTL also interact with age?

--how many putative causal SNPs were in the deleted 124 kb region upstream of POP5? Given the large size of this region, the evidence that this edit impacts POP5 regulation through the causal variant impacting TL is weakened.

Minor comments:

--abstract should mention the DNA source(s) for TL measurement. This is important so that the enrichment results are not misinterpreted.

--abstract should mention number of novel signals identified.

--Introduction: "... to identify 56 loci associated with human telomere length." This could be interpreted as 56 novel signals. please clarify.

--Please provide a table with sample sizes for each cohort. Supp table 1 is currently topped only. There are four other GWASs mentioned in the text.

--Fig 1 shows 8 genes in blue, so based on the figure alone, one cannot tell which are the 7 signals reported in the legend.

--Fig 2A needs horizontal axis labels that also provide a gene name, so the reader can match each signal to Fig 1.

Fig 2E. it is hard to see the SNPs in each of the confidence sets. Unclear how many SNPs in each set.

--Fig 3: do the blue and/or red points show higher levels of enrichment than the grey? Also, unclear if/how depletion is shown in this figure? what are the total counts of blue/red/grey/total? Same question for supp fig 3.

--Fig 3 title does not really reflect what's in panel B (only A)

--Fig 3; panel A needs a color code/key

--I recommend removing the word "ablate" and state more specifically the allele change.

--"A LeafCutter (Y. I. Li et al. 2018) plot of this splicing cluster demonstrated that individuals with more copies of the lead SNP at this locus increasingly excluded the fourteenth exon in RFWD3 (Figure 2D)." this statement needs reference a specific allele, not copies of "the SNP"

--"The rs2296312 locus was the single locus that met the filtering 469 pipeline criteria with a SNP x age interaction p-value = 2.599×10^{-6} (Figure 4A)." this analysis was restricted to regions identified in the meta-analysis. unclear if this interaction signal is the same as the meta-analysis signal. Do they co-localize?

--this heading sounds misleading "Blood and immune cells are a key cell type for telomere length". This would be TL measured in leukocytes specifically."

--in the methods, it would be helpful to indicated in what lab each of the experiments was done.

--Fig 5. Needs a clear message in the legend. Without going to the text, the message for panel A is unclear. The reader needs more help to quickly interpret panel A.

--Fig 5 B-C. The Legend should explain what the individual dots represent. Also, there appear to be more dots for some tissue categories than others? Does that impact the probability that a given tissue category shows evidence of enrichment?

--Fig 6: Is it possible to generate P-values with this type of data?

--Fig 6: I suggest slight realignment of the clone numbers and PD indicators on the figures. They are a bit off center, particularly for panel A, clone 7

--Fig 7, panel B. why are these SNPs with small P-values that are not in the credible set? Seems strange. And what are the small black boxes at the bottom?

--discussion, 1st paragraph. "We propose that blood and immune cells are the most relevant cell type for leukocyte telomere length GWAS validation experiments, but that these genes contribute to telomere length regulation across cellular contexts". Unclear which genes are being referred to in this sentence.

Reviewer #3 (Remarks to the Author):

In this manuscript, the authors conducted a trans-ancestry GWAS meta-analysis of human telomere length, wherein the authors identify blood and immune cells as the most pertinent cell types for investigating telomere length association signals, using enrichment analyses of chromatin state and cell-type heritability. Notably, the study identifies POP5 and KBTBD6 as novel human telomere length regulation genes. These findings are further validated through CRISPR/Cas9 experiments and telomere Southern blot analysis. Overall, the study provides a valuable contribution to our understanding of the genetic underpinnings of telomere length and its causal genes. However, I have several concerns and suggestions that I believe could further enhance the quality of the manuscript as follows:

Major:

1. The authors concluded that blood and immune cells were the most relevant cell type for telomere length association signals. Was this outcome influenced by investigating GWASs and eQTL studies within the context of blood and immune cells?
2. The authors employed a minor allele frequency (MAF) threshold of 0.1% for the meta-analysis; however, it is noteworthy that the Singapore cohort exclusively furnished GWAS summary statistics for variants considered common (MAF > 0.01). Not sure if the selection of 0.1% MAF will affect the meta-analysis performed.
3. The authors need address the suitability of using a fixed effects meta-analysis approach. While there exists a correlation among the measurements, including TelSeq, qPCR, and Luminex-based platform, the manuscript lacks a thorough discussion on how these correlations could potentially influence the meta-analysis results. The choice of employing a fixed effects model should be elaborated upon in light of these correlations and their potential impact on the overall conclusions drawn from the meta-analysis.
4. In contrast to the more stringent thresholds embraced by TOPMed, the authors opted for a relatively more lenient genome-wide significance threshold. An elucidation of the underlying rationale for this decision is imperative. It is also imperative to clarify the definition of novel signals within the context of the study, particularly when considering the varying GWAS significance thresholds across the studies. Equally vital is the question of whether certain "novel" loci underscored in this study satisfy the 5×10^{-8} threshold in the original GWAS but remained undisclosed due to the enforcement of the stricter threshold. To enhance reader comprehension, the inclusion of cohort-specific effect sizes and p-values in Supplementary Tables is recommended, affording a comprehensive evaluation of the specific contributions of different cohorts to the overall findings.
5. The extent of reproducibility for the reported sentinels in the GWAS meta-analysis warrants clarification. Additionally, an elaboration on the strategies employed by the authors to address potential heterogeneity across various studies is necessary. The approaches used to mitigate the impact of varying methodologies, sample sizes, and population characteristics on the consistency of findings among different cohorts should be expounded upon.
6. The authors asserted the identification of potential causal variants and genes influencing telomere length through fine-mapping analyses. However, in reality, the authors employed colocalization to pinpoint causal genes influencing telomere length. Utilizing fine-mapping to draw such conclusions is not suitable. Additionally, colocalization is not the sole method to detect causal genes, the authors are

encouraged to consider employing other methods such as TWAS and SMR to further validate their results.

7. The assumption in Coloc of a single shared causal variant between the eQTL and GWAS signal contradicts the multiple causal variants assumption of SuSiE. If the authors aim to showcase the existence of multiple causal variants within a GWAS signal, it would be prudent to consider alternative colocalization tools that do not rely on a singular causal assumption. Furthermore, the authors should consider implementing specific criteria for the utilization of PPH3 in the context of colocalization analysis.

8. It is essential for the authors to elucidate the rationale behind associating the signal from rs35510081 with TERC. As described by the authors, TERC is not in immediate proximity to rs35510081, nor is it colocalized with any eQTLs. The authors should support this attribution with references that establish a link between rs35510081 and TERC, thus bolstering the validity of their claim.

9. The enrichment analysis of TFBS may be performed using lead SNPs or best SNPs within the 95% credible set in addition to all SNPs within the credible set, as the outcomes may introduce bias due to the presence of multiple causal SNPs within each GWAS signal could be rare.

10. The authors are encouraged to provide comprehensive details regarding the GWAS analysis that integrates an age*genotype interaction term in the supplementary note. Furthermore, a thorough comparison between the original GWAS outcomes and the findings derived by the authors from the TOPMed dataset should be undertaken to underscore the importance of the interaction term. The original GWAS outcomes for rs2296312 locus should also be depicted in Figure 4A.

Minor:

1. What is the consideration for choosing different PPH4 threshold for GTEx, eQTLGen, and DICE to be colocalized. Moreover, the colocalization results for GTEx whole blood samples are better be listed as an independent category in Figure 2a.

2. The authors asserted that " SuSiE identified two credible sets for the signal led by rs35510081," did the authors imply that "SuSiE identified two SNPs within credible sets for the signal led by rs35510081"?

3. The authors should include an explanation of the criteria utilized for the selection of Bio genes in Supplementary Figure 2.

4. To enhance the reproducibility of the experiments, sequences of telomere probe and gRNAs should be included in the method section of telomere Southern blot analysis and CRISPR/Cas9 experiment.

5. The authors should provide a discussion regarding the factors contributing to the inability to identify 95% credible sets at 18 out of 56 loci.

6. It is recommended that the abstract include a presentation of the outcome where the overexpression of POP5 or KBTBD6 leads to an increase in telomere length.

REVIEWER COMMENTS

Reviewer #1 (Remarks to the Author):

In the study entitled “Validation of human telomere length trans-ancestry meta-analysis association signals identifies 2 POP5 and KBTBD6 as novel human telomere length regulation genes”, Keener et al. identified 56 genome-wide significant loci ($P < 5e-8$) associated with leukocyte telomere length including 7 novel signals through trans-ancestry meta-analysis. Five genes (OBFC1, PSMB4, CBX1, KBTBD6, and POP5) were selected for validation experiments and two were demonstrated to be involved in telomere length regulation. As stated in the Abstract, the noteworthy result is that their work “validated these putative novel genes to confirm the contribution of GWAS loci to telomere length regulation” through statistical analyses such as trans-ancestry meta-analysis, heritability enrichment analyses and validation experiments.

However, the major concern with the study is that the link between statistical results in the first few sections and the subsequent validation experiments look weak and it is difficult to see how the statistical analyses directly lead to the validation experiments and choice of genes for validation. For example, the authors state on page 13 “Therefore, we identified candidate genes where the lead SNP was predicted to increase gene expression. Of those we chose five genes that had one known protein coding sequence isoform, had strong colocalization analysis results, and had some known biology: OBFC1, PSMB4, CBX1, KBTBD6, and POP5 (Methods).” However, one of the validated genes POP5 claimed by the authors is not even mentioned once in the colocalization sections which the majority of the paper is focussed on. Another validated gene KBTBD6 was mentioned without defining strong colocalization (page 10). Furthermore, none of the genes above were identified in the novel signals yielded from trans-ancestry meta-analysis (blue dots in Figure 1). They are mentioned in Supplementary Note 1 but not all the five genes have the characteristics (isoform, colocalisation, known biology), according to this supplementary note.

It would be important for the authors to clarify this as currently, it is quite unclear how the validation experiments directly lead on from the work earlier in the study. Given that the discussion section begins with how the GWAS meta-analysis and fine-mapping work is a useful platform for further experimental validation, it is prudent for the authors to have clear and reproducible selection criteria about how these elements came together to provide putative targets for validation.

We thank the reviewer for pointing out this disconnect in the writing. To be transparent and clear about how genes were chosen for experimental validation we have added

Supplementary Note 2 and the corresponding code which walks through the filtering steps in detail. There, we discuss that after the association and colocalization pipelines to narrow down a list of candidates for followup, we also integrate knowledge of gene function and telomere biology to select a small set of genes for testing. Additionally, while our computational analysis informed the pipeline for selection of genes for experimental validation, some of the interesting computational results were not amenable to our specific experimental approaches, and in the scope of this single manuscript we only explored a limited mechanism and cell type. We observed strong computational support for several signals that did not qualify for our validation experiments but are highly interesting for future experimental work or integration with new QTL and disease data. For example, the results in Figure 2D which focus on *RFWD3* and splicing patterns detected by colocalization with splicing QTLs in GTEx, which did not qualify for experimental validation because the evidence suggested its effect was through splicing not expression. In the future researchers may orient their validation experiments to focus on molecular mechanisms such as splicing and highlighting these high confidence computational results may provide a starting point for that work.

Other comments:

1. The study identifies 2 - POP5 and KBTBD6 - as novel human telomere length regulation genes". However, POP5 and KBTBD6 are not among the 7 novel signals (Figure 1) by meta-GWAS. Is it appropriate for the authors to claim that they are (novelly) identified by trans-ancestry meta-analysis?

The reviewer is correct that the signals underlying both of these genes have been previously identified as associated with telomere length, however, our study is the first to experimentally validate a role for these genes in human telomere length regulation. We interpret this situation as a novel role in telomere length regulation. We have rephrased the text in the abstract to address this potential point of confusion.

2. The authors outlined three criteria for identifying candidate genes for screening validation experiments in the Methods (page 22). More information is needed such as how a meta-analysis signal or the strongest colocalization was defined. This information may have been provided somewhere else in the manuscript/supplementary file. But it would be more clear if a summarising table (or descriptions of key details e.g. meta-analysis P values, colocalization definition) can be provided in the Results section accordingly, which would also serve as a benchmark for future studies adopting a similar strategy. Similarly, 'The strongest enrichment of several active chromatin states was observed in blood and T-cell samples' (page 13). Which threshold value was used for the 'strongest enrichment'? Threshold information ought to be provided wherever applicable.

To improve clarity, guidance, and transparency about our validation experiments, we have expanded our explanation in the Results/Overexpression of *POP5* and *KBTBD6* increases telomere length in HeLa-FRT cells section (paragraph 1) added a section in our Methods (Identifying candidate genes for overexpression experiments) describing in greater detail how we chose genes for our validation experiments and the logic behind thresholds used. In addition, we detailed Supplementary Note 2, which conveys the data used to make these decisions.

The reviewer also specifically requested a more quantitative conclusion for the Roadmap chromatin state enrichment analysis. To quantitatively examine which cell types had the strongest enrichment for each chromatin state we used Fisher's method to calculate combined chi-squared statistic across the cell type samples (ex. Blood and T cells) for each chromatin state (ex. 1_TssA). The strongest enrichment was defined as the cell type sample group with the most significant p-value compared across all cell type sample groups. We observed that the Blood and T cell group was had the strongest enrichment across the most active chromatin states and we have included this result in Supplemental Figure 5C, refer to it in the Results/Blood and immune cells are a key cell type for leukocyte telomere length section, and included a description of this analysis in our Methods (Enrichment of meta-analysis signals in chromatin states).

3. The authors state the following criteria 'the lead variant at the meta-analysis signal was required to be associated with increased gene expression'. Does this mean that a lead variant should also be an eQTL? How can a variant be associated with a direction of gene expression? Shouldn't it be modelled as a specific allele (e.g., positive/negative-effect allele) of trait A (i.e., telomere length) that promotes/reduces trait B (i.e., gene expression)? There are other similar descriptions elsewhere in the manuscript that should also be rephrased/clarified. Again, a threshold for such associations with gene expression ought to be provided.

We have adjusted the description of this requirement for candidate genes for our validation experiments. Because we planned to overexpress the candidate genes, we required that the lead SNP at the meta-analysis signal was also an eSNP and we required that the GWAS tested alternate allele have a positive effect estimate (that with increasing alternate allele copies, expression increased). The reviewer also requested that we convey a threshold for such associations and we have added that we required that the lead SNP be a significant eSNP (FDR < 0.05) in GTEx. We made this adjustment in both the Methods/Identifying candidate genes for overexpression experiments section, Results/Overexpression of *POP5* and *KBTBD6* increases telomere length in HeLa-FRT cells section, and in Supplemental Note 2.

4. I would suggest simplifying the Results especially for the statistical analyses – there appears to be extended description and discussions for colocalization analyses (4 sections) that are difficult to follow.

We hope that our manuscript will be a useful guide to researchers who seek to validate GWAS findings. Because colocalization analysis is a highly used tool for this goal, we think it is important to provide guidance on how colocalization results should be interpreted. To that end, we think that maintaining our sections on defining colocalization and explaining some key limitations, showcasing the value of colocalization with splicing QTLs, and how to deal with challenging circumstances where colocalization results give conflicting results, are important to this manuscript.

5. Is the finding that ‘TCL1A 95% credible set SNPs are more strongly associated with telomere length in older individuals’ relevant to the context? Otherwise it appears like a bit of a distraction.

A key theme of this manuscript is different approaches to identifying the molecular mechanisms underlying GWAS signals. We have adjusted the transition to this section by explaining that certain signals may have a temporal component and have a greater effect in younger or older individuals. This is especially relevant for telomere length because age accounts for a large proportion of telomere length variation in the human population.

6. Regarding the blood/T-cell enrichment (page 13), I would suggest that the authors rephrase the relevant content to improve readability. For instance, explanation of ‘the chromHMM model is a predicted state’. Furthermore, what was the ‘primary data for specific chromatin marks’?

We have added clarification sentences to address both of these questions in the blood/T-cell enrichment section.

7. In the S-LDSC analysis detecting heritability enrichment (page 13), shouldn’t the cell type enrichment in blood cells be under expectation as the ‘Telomere length was estimated from blood leukocytes’?

The reviewer is correct, this was not a surprising finding since telomere length was estimated from blood leukocytes. We have added transparency to our description of these findings in the Discussion and emphasized that the samples were from blood leukocytes in the Results/Blood and immune cells are a key cell type for leukocyte telomere length section.

8. The authors found that "blood/immune cell category was the only category that was significantly enriched" (page 13). However, at the beginning of that section, the authors cited a published finding that "telomere length regulation is relevant in many different cell types to different extent" (page 12). Is the finding in the present study contradictory to the published one? How would the authors explain this?

Thank you for the opportunity to clarify this point. We have updated the text to make it clear that while we only detected significant enrichment of our signals in the blood/immune cells category, this could be due to the fact that our samples were from blood/immune cells and it could be that with higher powered summary statistics or higher quality chromatin or expression data, potentially from additional cell types not represented here, significant enrichment of other cell types would be detected (Discussion).

9. In Figure 5B, does it suggest that the top enrichment signal (above the dash line) comes from the gene expression data in spleen cells, and others were not significant? What P-value does the dash line indicate? Are the labels on x-axis tissues or cell-types?

Yes, the only significant result from the gene expression data was from Spleen which Finucane et al. 2018 categorized as part of Blood and Immune cells. The dashed line represents the significance threshold of $FDR < 0.05$ at $-\log_{10}(p\text{-value}) = 2.75$. The labels on the x-axis are tissue categories and each data point is a cell type. This information has been added to the figure legend to improve clarity.

10. In page 13, "HeLa cells are not derived from blood or immune cells but are highly tractable for this screening stage of the validation experiments". This seems illogical and the choice of HeLa cells for validation seems to call into question the importance and relevance of the cell-type enrichment results?

From the outset of our validation experiments we planned a two stage approach where first we would dramatically overexpress candidate genes and look for an effect on telomere length and second a precise experiment where we would aim to demonstrate that the putative causal variant(s) affect the putative causal gene. As part of our filtering criteria for candidate genes we prioritized genes that were widely expressed across cellular contexts (Supplemental Note 2, Stage 4) because we considered that genes critical for telomere length maintenance through more direct mechanisms are likely going to be expressed consistently across different cellular contexts since telomere length maintenance must occur in all cells. This argument may not apply to mechanisms that function solely to directly regulate telomerase, which has cell-type specific activity, but given the candidate gene list we generated and what was known about these genes, we thought it unlikely that any of them would directly regulate telomerase. Furthermore, our overexpression experiment does not rely on any potential cell-type specific regulators such as transcription

factors. Therefore, we argue that it is reasonable to use HeLa cells for this first stage in order to improve technical aspects of the experiment. In comparison, the second stage of the validation experiments may impact cell-type specific regulatory mechanisms, for instance by removing a cell-type specific transcription factor. Therefore, we conducted the CRISPR/Cas9 editing experiments in K562 blood-cell derived cells.

11. Brief discussion of the potential clinical relevance of the study's findings should be included.

We have added a paragraph on this topic in the discussion.

Reviewer #2 (Remarks to the Author):

This meta-analysis of several prior GWAS of TL reports 56 signals for leukocyte TL, 7 of which are novel. This paper also characterizes mechanisms using QTL data and conducts experiments to identify causal genes and regulatory elements for two TL association signals (POP5 and KBTBD6). This paper is an important contribution to the TL literature, but I have several concerns below that can hopefully be addressed.

We thank the reviewer for their positive comments on the manuscript and their attention to both the figures and figure legends.

Major Comments:

--"...overexpression of KBTBD6 or POP5 showed a clear telomere length increase over increased cell division, concordant with the expectation from our meta-analysis (Figure 6)." Looking at Figure 6 (B and D), it does not appear that this statement is accurate. I see decreases in TL at the third time-point for KBTBD6 clone 7 and for POP5 clones 5 and 6. I do see increases for the second time point (but not the third).

The reviewer's interpretation of the data is accurate and we have adjusted our description of this figure in the Results/Overexpression of *POP5* and *KBTBD6* increases telomere length in HeLa-FRT cells section. We have also added an explanation as to why telomere length plateauing is not surprising for this type of experiment and has precedent in the literature for canonical telomere length regulation genes.

--when determining what signals are novel, what prior studies were included? Does this include the UK Biobank GWAS?

We thank the reviewer for pointing out the lack of clarity here. We have added a paragraph in the Methods/ Meta-analysis section that specifies how we determined whether a signal was

novel and which previously published telomere length GWAS were included in that analysis. We did include the UK Biobank GWAS.

--Given UKB results are not included in this meta-analysis, is it possible to replicate the novel signals detected here using the UKB summary statistics?

We thank the reviewer for this idea. We conducted a replication analysis using the UK Biobank summary statistics and now include a description of that in our Results/ Multi-ancestry meta-analysis of leukocyte telomere length identifies 5 novel signals section and Methods/ Replication analysis section. Of our five novel signals, four lead SNPs were examined in the UKB Biobank telomere length GWAS and we were able to replicate two of our novel signals in that dataset.

--Replication would also help for the POP5 signal, which has a borderline P-value. Has POP5 been reported previously? Given the importance of POP5 for this paper, more is needed to convince readers the association is real (and not a false positive).

While previous studies did not attribute association signals to *POP5*, the highest powered, published telomere length GWAS from the UK Biobank (Codd et al. 2021) detected a genome-wide significant signal nearby, and colocalization between the UK Biobank statistics at this locus and our association statistics for the signal we attribute to *POP5* is strong (PPH3 = 0.229, PPH4 = 0.77). Furthermore, colocalization analysis between the UK Biobank GWAS summary statistics and GTEx eQTLs replicated our colocalization findings for this signal. In both cases the highest PPH4 was with an eQTL for *GATC* (PPH3 = 0.199, PPH4 = 0.801 in UKBB, PPH3 = 0.145, PPH4 = 0.854 in our study). However, both studies also showed similar colocalization with an eQTL for *POP5* (PPH3 = 0.391, PPH4 = 0.532 in UKBB, PPH3 = 0.351, PPH4 = 0.565 in our study). We have added the replication of these colocalization analysis findings to Supplemental Note 2.

In the absence of experimental validation, we would suggest that *GATC* is the more likely candidate based solely on colocalization analysis. However, a theme of this manuscript is communicating the importance of weighing all available information for putative candidate genes. It was recently reported that another Ribonuclease P/MRP complex member, POP1, interacts with human telomerase RNA (Zhu et al. 2023). In addition, previous work in *S. cerevisiae* demonstrated a role for specific components of the homologous complex in telomerase holoenzyme complex regulation (Laterreur et al. 2018). We describe these publications in our Discussion. Based on the previous publications related to POP family of proteins and telomere length biology, we chose to validate *POP5* finding experimentally. Our experimental validation experiments, particularly the overexpression experiment, demonstrated that POP5 affects telomere length in human cells.

--In Fig 1, POP5 (and a few other regions) do not appear to cross the threshold, so readers will wonder why these are labeled as TL signals. Some clarification is needed.

We thank the reviewer for catching this detail. There was an error in the graphing code that placed the significance threshold too high. This has been corrected in the updated version of Figure 1.

--“Telomere length was estimated from blood leukocytes in all samples; it is possible that this boosted the strength of blood and immune cell enrichment in our analyses.” This is highly likely in my view. Also, this paragraph includes no mention of the fact that telomere maintenance by telomerase occurs primarily in stem and progenitor cells, not the differentiated cell/tissue types that make up most of GTEx, ENCODE, etc.. This seems very relevant to the 1st paragraph of the discussion.

We have added the fact that telomerase is only detectably active in progenitor and stem cells in addition to blood and immune cells in the Results/Blood and immune cells are a key cell type for leukocyte telomere length section. We have also added the fact that the enrichment analyses rely on terminally differentiated cell types, which largely do not have active telomerase. However, we also added the point that the majority of the association signals likely underlie telomere length regulations that are independent of directly regulating telomerase and likely represent more indirect telomere length regulation mechanisms that are relevant across cell types (Results/Blood and immune cells are a key cell type for leukocyte telomere length section).

--Discussion: “Together, these results suggest that the role of the POP proteins also play a role in human telomerase regulation” is telomerase active in the cell type(s) used for experimental validation?

Yes, telomerase is active in HeLa cells where the overexpression experiments were conducted (PMID: 16955216).

--is it possible to attempt replication in an independent data source of the SNP-by-Age interaction for *TCL1A*? And does the eQTL also interact with age?

We are not able to attempt replication of the SNP x Age interaction for *TCL1A*. We evaluated whether the eQTL has an interaction with age but it does not. However, *TCL1A* expression has previously been reported to decrease with age in whole blood (PMID: 32913074).

--how many putative causal SNPs were in the deleted 124 kb region upstream of POP5?
Given the large size of this region, the evidence that this edit impacts POP5 regulation through the causal variant impacting TL is weakened.

Putative causal variants are identified through 95% credible set estimation. We were unable to estimate a 95% credible set (or lower threshold) for the *POP5* locus using either SuSiE or CAVIAR credible set prediction. This was likely because the signal is not well powered. Therefore, we are unable to estimate the number of putative causal SNPs. We attempted to leverage orthogonal data to narrow down the region computationally by intersecting top SNPs ($r^2 > 0.9$ with the lead SNP and $p\text{-value} < 1 \times 10^{-6}$) with ATAC-seq, Hi-C, and chromatin ChIP-seq data from blood samples, but were unable to form a consensus. While this experiment was not as precise as would be ideal, we do not attempt to make claims about specific variants being the causal variant, however to the best of our ability, our work suggests that the causal variant is in this region.

Minor comments:

--abstract should mention the DNA source(s) for TL measurement. This is important so that the enrichment results are not misinterpreted.

We have added this information to the abstract.

--abstract should mention number of novel signals identified.

We have added this information to the abstract.

--Introduction: "... to identify 56 loci associated with human telomere length." This could be interpreted as 56 novel signals. please clarify.

We have added this information to the introduction.

--Please provide a table with sample sizes for each cohort. Supp table 1 is currently topped only. There are four other GWASs mentioned in the text.

We have added this information to supplemental table 1.

--Fig 1 shows 8 genes in blue, so based on the figure alone, one cannot tell which are the 7 signals reported in the legend.

We have corrected this error.

--Fig 2A needs horizontal axis labels that also provide a gene name, so the reader can match each signal to Fig 1.

We added the chromosome arm and segment number for each lead SNP to facilitate comparison of Figure 2A to Figure 1. We did not want to add the attributed gene name in Figure 2A out of concern that a reader would think that the colocalization results described in Figure 2A all supported the attributed gene.

Fig 2E. it is hard to see the SNPs in each of the confidence sets. Unclear how many SNPs in each set.

We reduced the x-axis to more predominantly show the association signal, but were unable to make the credible set SNPs countable by eye without excluding significantly associated SNPs in the signal. We have added the number of SNPs in each credible set to the figure legend. In addition, this information is available in Supplementary Table 7 and the figure legend also directs readers there for a list of snpIDs (chr:pos:ref:alt) of SNPs in the credible sets.

--Fig 3: do the blue and/or red points show higher levels of enrichment than the grey? Also, unclear if/how depletion is shown in this figure? what are the total counts of blue/red/grey/total? Same question for supp fig 3.

Figure 3A shows the enrichment of transcription factor binding sites SNPs in the meta-analysis signal 95% credible sets. We highlighted transcription factors with known roles in regulating expression of telomere length regulation genes in red and transcription factors with known roles in the alternative lengthening of telomeres (ALT) pathway. In the literature the focus of many of these transcription factors has been limited to studying the regulation of *TERT* expression. Many of the red and blue labelled points are enriched, and we suggest that researchers should consider examining candidate telomere length regulation genes nominated by association studies for transcriptional regulation by transcription factors known to regulate some telomere length regulation genes as the enrichment of these transcription factors suggest they may bind at multiple association signals.

There are also many transcription factors with no known role in regulating telomere length regulation genes (grey points) that are enriched. We do not claim that transcription factors with known roles related to telomere length (red and blue) are more strongly enriched than those with no known role (grey). It is possible that some of these very strongly enriched transcription factors have unexamined roles in telomere length regulation and we encourage readers to examine them more closely.

We have added legends and the total counts of red, blue, and grey dots to the figure legends for Figure 3A and Supplemental Figure 3.

--Fig 3 title does not really reflect what's in panel B (only A)

We have updated the figure title.

--Fig 3; panel A needs a color code/key

We have added a color code/key to Figure 3A and Supplementary Figure 3A-C.

--I recommend removing the word "ablate" and state more specifically the allele change.

We have changed the word "ablate" to "alter" and have specified the allele change.

--"A LeafCutter (Y. I. Li et al. 2018) plot of this splicing cluster demonstrated that individuals with more copies of the lead SNP at this locus increasingly excluded the fourteenth exon in RFW3 (Figure 2D)." this statement needs reference a specific allele, not copies of "the SNP"

We have updated the text to be specify a particular allele and genotype.

--"The rs2296312 locus was the single locus that met the filtering 469 pipeline criteria with a SNP x age interaction p-value = 2.599×10^{-6} (Figure 4A)." this analysis was restricted to regions identified in the meta-analysis. unclear if this interaction signal is the same as the meta-analysis signal. Do they co-localize?

The SNP x age interaction signals that clear a suggestive threshold are all genome-wide significant in the meta-analysis signal attributed to *TCL1A*. Because the SNP x age interaction signal was only suggestive, colocalization analysis cannot accurately evaluate whether these two signals share a common causal variant.

--this heading sounds misleading "Blood and immune cells are a key cell type for telomere length". This would be TL measured in leukocytes specifically."

We have updated the section title to, "Blood and immune cells are a key cell type for leukocyte telomere length".

--in the methods, it would be helpful to indicated in what lab each of the experiments was done.

We have added this information in the Author Contribution sections of this manuscript.

--Fig 5. Needs a clear message in the legend. Without going to the text, the message for panel A is unclear. The reader needs more help to quickly interpret panel A.

We have updated the figure legend to guide the reader in their interpretation of panel A.

--Fig 5 B-C. The Legend should explain what the individual dots represent. Also, there appear to be more dots for some tissue categories than others? Does that impact the probability that a given tissue category shows evidence of enrichment?

We have updated the figure legend to explain what individual dots represent. It's true that some of the tissue categories have more samples than others, and some of the cell types are better powered than others, as described in Finucane et al. 2018. How this impacts the analysis depends on the variation in the signal across cell types within the tissue categories. Some of the samples with the strongest signal in the "Other" tissue category are fetal cell types, which may become significant as new expression or chromatin data become available for use in these analysis.

--Fig 6: Is it possible to generate P-values with this type of data?

Unfortunately not.

--Fig 6: I suggest slight realignment of the clone numbers and PD indicators on the figures. They are a bit off center, particularly for panel A, clone 7

We have corrected the alignment on all panels in Figure 6 as needed.

--Fig 7, panel B. why are these SNPs with small P-values that are not in the credible set? Seems strange. And what are the small black boxes at the bottom?

Some SNPs with small p-values were modeled by SuSiE to have close to 0 posterior inclusion probability. This is sometimes observed with the SuSiE model as it estimates the smallest credible set (in terms of the number of genetic variants) that accounts for the association signal. The small black boxes at the bottom indicate a second predicted credible set that modeled to be independent of the first credible set. This information was missing in the Figure 7 legend and we have added it.

--discussion, 1st paragraph. "We propose that blood and immune cells are the most relevant cell type for leukocyte telomere length GWAS validation experiments, but that these genes contribute to telomere length regulation across cellular contexts". Unclear which genes are being referred to in this sentence.

We have updated the indicated text to improve clarity.

Reviewer #3 (Remarks to the Author):

In this manuscript, the authors conducted a trans-ancestry GWAS meta-analysis of human telomere length, wherein the authors identify blood and immune cells as the most pertinent cell types for investigating telomere length association signals, using enrichment analyses of chromatin state and cell-type heritability. Notably, the study identifies POP5 and KBTBD6 as novel human telomere length regulation genes. These findings are further validated through CRISPR/Cas9 experiments and telomere Southern blot analysis. Overall, the study provides a valuable contribution to our understanding of the genetic underpinnings of telomere length and its causal genes. However, I have several concerns and suggestions that I believe could further enhance the quality of the manuscript as follows:

We thank the reviewer for their insight and particular attention to how the meta-analysis was conducted. We respond to their concerns below.

Major:

1. The authors concluded that blood and immune cells were the most relevant cell type for telomere length association signals. Was this outcome influenced by investigating GWASs and eQTL studies within the context of blood and immune cells?

The reviewer raises a good point, the fact that telomere length was estimated from blood leukocytes may have impacted both the chromatin state enrichment and S-LDSC analyses. We acknowledge this point in our Discussion (paragraph 1). We have adjusted how we describe these experiments to be more clear about this point in both the Results/ Blood and immune cells are a key cell type for leukocyte telomere length and in the Discussion. This outcome was not influenced by examining eQTLs derived from blood and immune cells, although we expanded our analysis to the eQTLGen and DICE eQTL datasets upon observing the outcome of the chromatin state enrichment and S-LDSC analyses.

2. The authors employed a minor allele frequency (MAF) threshold of 0.1% for the meta-analysis; however, it is noteworthy that the Singapore cohort exclusively furnished

GWAS summary statistics for variants considered common (MAF > 0.01). Not sure if the selection of 0.1% MAF will affect the meta-analysis performed.

As the reviewer states, the Singaporean Chinese study (Dorajoo et al. 2019) used a threshold of MAF > 0.01 and used genomic imputation. The Bangladeshi study (Delgado et al. 2018) used a threshold of MAF > 0.01 and the European study (Li et al. 2020) used a threshold of MAF > 0.01 for HapMap3 SNPs and MAF > 0.05 for 1000 Genomes European-imputed SNPs and both studies also used genomic imputation. The TOPMed cohorts used whole genome sequencing across all samples with no genomic imputation, therefore we retained all available genetic variant information as we hoped it would improve our fine-mapping analyses. Three genome-wide significant meta-analysis signals have a lead SNP with meta-analyzed effect allele frequency < 0.01: rs542948485, rs73687065, rs958919990.

In two cases the most significantly associated SNPs have EAF < 0.01, but the signal overall consists of SNPs with EAF \geq 0.01. The signal led by rs958919990 solely consists of SNPs with EAF < 0.01, demonstrating that including these lower allele frequency SNPs were valuable to include in the meta-analysis. Interested readers may investigate characteristics of the SNPs as we provide our summary statistics with this manuscript.

3. The authors need address the suitability of using a fixed effects meta-analysis approach. While there exists a correlation among the measurements, including TelSeq, qPCR, and Luminex-based platform, the manuscript lacks a thorough discussion on how these correlations could potentially influence the meta-analysis results. The choice of employing a fixed effects model should be elaborated upon in light of these correlations and their potential impact on the overall conclusions drawn from the meta-analysis.

We agree that the choice of a fixed effects as opposed to a random effects meta-analysis is a critical choice. Despite reasonable correlation between telomere length estimates, we chose a more conservative approach of using a random effects meta-analysis, as described in our Methods/Meta-analysis section.

4. In contrast to the more stringent thresholds embraced by TOPMed, the authors opted for a relatively more lenient genome-wide significance threshold. An elucidation of the underlying rationale for this decision is imperative. It is also imperative to clarify the definition of novel signals within the context of the study, particularly when considering the varying GWAS significance thresholds across the studies. Equally vital is the question of whether certain "novel" loci underscored in this study satisfy the 5×10^{-8} threshold in the original GWAS but remained undisclosed due to the enforcement of the stricter threshold. To enhance reader comprehension, the inclusion of cohort-specific effect sizes and p-values in Supplementary Tables is recommended, affording a comprehensive evaluation of the specific contributions of different cohorts to the overall findings.

We have generated a new Supplementary Table 3 which provides comparisons of p-values, effect size estimates, and the standard error of the effect size estimates for meta-analysis lead SNPs across the cohorts used in this analysis. In addition, we provide data from the TOPMed pooled GWAS for comparison, which jointly examined all TOPMed ancestry groups. None of the signals we report as novel had p-values $< 5 \times 10^{-8}$ in the original studies. Of our novel signals, the signal led by rs958919990 in the meta-analysis, was genome-wide significant in the TOPMed European dataset (lead SNP p-value = 7.07×10^{-9}), however, it did not meet the MAF threshold (MAF < 0.01) and was not reported.

A second novel signal, led by rs12241155 in the meta-analysis, had a p-value = 1.18×10^{-10} in the TOPMed pooled analysis, but was not reported as a signal. Note that our meta-analysis does not use the TOPMed pooled analysis as input, but this is relevant for considering whether this signal should be considered novel. After consulting with the lead and corresponding authors on that study (Taub et al. PMID 35530816), we learned that this was because there was evidence

in their work that the signal is not conditionally independent (provided figure). In the TOPMed pooled analysis the signal with the strongest association was attributed to OBFC1 and led by rs9420907 (vertical red line). The neighboring signal, led by rs12241155 in the meta-analysis, is indicated with a vertical blue line. After conditioning upon rs9420907, the signal led by rs12241155 in the meta-analysis was no longer genome-wide significant ($p < 5e-9$, horizontal blue line). Therefore, this signal was not included that manuscript. In light of this, we no longer consider the signal led by rs12241155 in the meta-analysis to be novel and we address this in the Methods/Meta-analysis section.

5. The extent of reproducibility for the reported sentinels in the GWAS meta-analysis warrants clarification. Additionally, an elaboration on the strategies employed by the authors to address potential heterogeneity across various studies is necessary. The approaches used to mitigate the impact of varying methodologies, sample sizes, and population characteristics on the consistency of findings among different cohorts should be expounded upon.

We agree that the potential impact of varying methodologies, sample sizes, and population characteristics may impact the findings. To address this we used a GWAMA random-effects meta-analysis model. We also examined the extent of heterogeneity as measured by the Cochran's q statistic and the I^2 statistic. We report the values for both measures in Supplementary Table 2 for lead SNPs. None of our lead SNPs had significant heterogeneity by either measure. We have elaborated on this in our Results/Multi-ancestry meta-analysis of leukocyte telomere length identifies 6 novel signals section.

6. The authors asserted the identification of potential causal variants and genes influencing telomere length through fine-mapping analyses. However, in reality, the authors employed colocalization to pinpoint causal genes influencing telomere length. Utilizing fine-mapping to draw such conclusions is not suitable. Additionally, colocalization is not the sole method to detect causal genes, the authors are encouraged to consider employing other methods such as TWAS and SMR to further validate their results.

The reviewer is correct that orthogonal methods exist and their inclusion improves our work. We have added a TWAS analysis using a model trained on expression data from whole blood in the Young Finns Cohort. This analysis identified 19 associated genes, 9 of which were not within 100 Mb of a genome-wide significant locus, demonstrating the power of TWAS to identify genes associated with telomere length. Of the 10 genes near meta-analysis signals, 7 genes were concordant with genes nominated by colocalization analysis. We have added a paragraph discussing these results in the Results/Fine-mapping analysis nominate putative causal variants and genes affecting telomere length/Colocalization analysis and TWAS suggests genes underlying association signals

and elaborate on them in Supplemental Note 1, where we discuss the rationale for the gene assignment for each meta-analysis signal.

7. The assumption in Coloc of a single shared causal variant between the eQTL and GWAS signal contradicts the multiple causal variants assumption of SuSiE. If the authors aim to showcase the existence of multiple causal variants within a GWAS signal, it would be prudent to consider alternative colocalization tools that do not rely on a singular causal assumption. Furthermore, the authors should consider implementing specific criteria for the utilization of PPH3 in the context of colocalization analysis.

We chose to conduct our colocalization analysis using the coloc method as it is widely used and more computationally efficient than similar methods, such as eCAVIAR. eCAVIAR calculates a CLPP score for each GWAS variant with eQTLs and is able to handle cases where there are multiple causal variants at the locus. Putative target genes are ranked according to the aggregate CLPP scores, therefore, like coloc, eCAVIAR reports the likelihood that an association signal is linked to a particular gene. The eCAVIAR publication did not evaluate cases where there were a differing number of causal variants between the GWAS association signal and the QTL. This may also impact the performance of eCAVIAR as the QTL datasets are not as well powered as the GWAS. Furthermore, many of the non-primary meta-analysis association signals in our study are below genome-wide significance ($p < 5 \times 10^{-8}$) making them less amenable to colocalization. The eCAVIAR publication did examine the impact of linkage disequilibrium on eCAVIAR performance and found that in the case of complex linkage disequilibrium structure the CLPP may be small even when there is true colocalization (PMID: 27866706). Many of the loci where SuSiE 95% credible set prediction suggested there are multiple causal variants have complex linkage disequilibrium structures. Furthermore, the linkage disequilibrium structure is distinct between our multi-ancestry meta-analysis and the QTL datasets, which are largely European, this would likely reduce the additional insights analysis with eCAVIAR would make. However, additional methods exist that may better handle these complex scenarios, such as CAFEH (PMID: 35085493), a method developed by our group that leverages multi-trait or multi-tissue data to identify shared causal variants across contexts. Exploring such analyses would be of interest in future work.

The reviewer also suggested that we implement specific criteria for PPH3 in our colocalization analysis. As the posterior probabilities of the five potential coloc outcomes must sum to 1, placing a threshold of $PPH4 > 0.7$ implicitly enforces a threshold on the other posterior probabilities and the maximum allowed PPH3 is 0.3. Some studies do utilize a threshold on $PPH4/(PPH4 + PPH3)$, but this is more permissive than our approach, and we chose to maintain our threshold.

8. It is essential for the authors to elucidate the rationale behind associating the signal from rs35510081 with TERC. As described by the authors, TERC is not in immediate proximity to rs35510081, nor is it colocalized with any eQTLs. The authors should support this attribution with references that establish a link between rs35510081 and TERC, thus bolstering the validity of their claim.

None of our QTL datasets have QTLs for *TERC*, likely because it has very low expression across cell types. However, *TERC* is irrefutably a key gene for telomere length regulation as it is a core component of telomerase (PMID: 7544491) and these genetic variants may affect *TERC* through additional molecular mechanisms outside of expression and splicing.

We evaluated novelty of our meta-analysis signals as a signal that was not within 1Mb of a previously reported lead SNP in a telomere length GWAS. There are three variants associated with telomere length and attributed to *TERC* within 1Mb of the lead variant in the meta-analysis signal that we attribute to *TERC* (see table below). We examined the linkage disequilibrium between rs35510081 and these variants and found them to be in low to moderate linkage disequilibrium with rs35510081 using a multi-ancestry reference panel from TOPMed, further supporting that it is reasonable to attribute our meta-analysis signal led by rs35510081 to *TERC*. We have added this point about distance and linkage disequilibrium with known telomere length associated and *TERC* attributed signals to our Results/Credible set analysis suggests that some loci consist of multiple independent causal variants which regulate the same gene in different contexts section.

Study	Study PMID	Lead SNP attributed to TERC	r2 with our lead SNP attributed to TERC
Dorajoo et al.	31171785	rs2293607	0.62
Taub et al.	35530816		
Codd et al.	20139977	rs12696304	0.20
Prescott et al.	21573004		
Taub et al.	35530816		
Delgado et al.	29151059	rs12638862	0.49
Taub et al.	35530816		

9. The enrichment analysis of TFBS may be performed using lead SNPs or best SNPs within the 95% credible set in addition to all SNPs within the credible set, as the outcomes may

introduce bias due to the presence of multiple causal SNPs within each GWAS signal could be rare.

The reviewer is correct that differences in the size of the 95% credible sets may influence the enrichment analysis. While this analysis is calibrated to a set of background variants, it is possible that some loci have a stronger effect and have a greater influence on the enrichment observed. We have added this caveat to our description of this analysis in the Results/Meta-analysis signals are enriched for transcription factor binding sites of transcription factors with roles in telomere length regulation section.

10. The authors are encouraged to provide comprehensive details regarding the GWAS analysis that integrates an age*genotype interaction term in the supplementary note. Furthermore, a thorough comparison between the original GWAS outcomes and the findings derived by the authors from the TOPMed dataset should be undertaken to underscore the importance of the interaction term. The original GWAS outcomes for rs2296312 locus should also be depicted in Figure 4A.

To address the reviewer's points we have added six panels to the manuscript. We added a Manhattan plot for the original GWAS outcome for the rs2296312 to Figure 4 as requested (now Figure 4A). Because Supplemental Note 1 focuses on evidence suggesting putative causal genes underlying each signal, we thought introducing the age x genotype data there may confuse readers. Instead, we have created five new panels in Supplementary Figure 4 which provide a summary of results from the GWAS with an age x genotype interaction term. Supplementary Figure 4C is a Miami plot comparing the summary statistics for both datasets. Supplementary Figure 4D is a scatterplot comparing the strength of association of peaks with a lead SNP that is at least suggestive ($p < 5 \times 10^{-5}$) in either the original GWAS or the GWAS with the age x genotype interaction term. We provide this panel to demonstrate that many signals that are significant in the original GWAS do not have significant age x genotype interaction effects. We highlight three lead SNPs (rs8012195, rs2515349, and rs585168) which had at least suggestive significance in both the original TOPMed pooled GWAS and the GWAS with an age x genotype interaction term (colored points in Supplementary Figure 4D). rs8012195 is the lead variant near *TCL1A* and is the focus of Figure 4. Supplementary Figure 4E shows Manhattan plots around rs2515349 and shows that that this lead SNP is not part of an association signal, rather a lone significant SNP in the GWAS with a genotype x age interaction term, in the original TOPMed pooled GWAS, and in the meta-analysis. Supplementary Figure 4F shows Manhattan plots for rs585168, which is near the signal we attribute to *MIR223HG*. Supplementary Figure 4G shows a forest plot for the effect size estimate and effect size standard error for rs585168. Because the effect size estimate does not linearly trend with age across the age-stratified GWAS, we did not focus on it in the main text, but with this reviewer's encouragement, now provide it in the

supplement for interested readers. We discuss the data in these new panels in Results/*TCL1A* 95% credible est SNPs are more strongly associated with telomere length in older individuals section.

Minor:

1. What is the consideration for choosing different PPH4 threshold for GTEx, eQTLGen, and DICE to be colocalized. Moreover, the colocalization results for GTEx whole blood samples are better be listed as an independent category in Figure 2a.

Colocalization analysis is sensitive to the strength of the association signal and the strength of the QTL. Because the DICE database has a low sample size, overall the eQTLs have reduced association strength. On the other hand, GTEx and eQTLGen are well powered. Therefore, we allowed a reduced threshold of PPH4 > 0.5 for DICE to be considered evidence of colocalization. However, we were careful to note that none of the genes attributed to meta-analysis association signals are supported by DICE colocalization data alone. Rather, DICE colocalization results support other colocalization results in several cases and provide insight into potential cell types where this signal might particularly relevant (ex.rs59922886 in Supplemental Note 1).

2. The authors asserted that " SuSiE identified two credible sets for the signal led by rs35510081," did the authors imply that "SuSiE identified two SNPs within credible sets for the signal led by rs35510081"?

The signal led by rs35510081 was predicted to have two independent causal variants, with ten SNPs in the first credible set and four SNPs in the second credible set. We have updated the text to clarify this point.

3. The authors should include an explanation of the criteria utilized for the selection of Bio genes in Supplementary Figure 2.

We agree that this was an important addition. We have expanded our on description in the main text in the Results/Overexpression of POP5 and KBTBD6 increases telomere length in HeLa-FRT cells, created a section for this in the Methods/Identifying candidate genes for overexpression experiments, and generated Supplemental Note 2 which walks through this process in extensive detail.

4. To enhance the reproducibility of the experiments, sequences of telomere probe and gRNAs should be included in the method section of telomere Southern blot analysis and CRISPR/Cas9 experiment.

The sequences of the telomere probe and gRNAs are specified in Supplementary Table 14. We have elaborated on how the telomere Southern blot probe is generated in our Methods section (see Telomere Southern blot section). In addition, we provide the plasmid maps that contain these sequences to further improve reproducibility. These plasmid maps are included in our Zenodo data upload.

5. The authors should provide a discussion regarding the factors contributing to the inability to identify 95% credible sets at 18 out of 56 loci.

We examined this and attribute this to the relative strength of the association. Signals with weaker association tended not to have a SuSiE 95% credible set. We have added an explanation of this in the Results/Credible set analysis suggests that some loci consist of multiple independent causal variants which regulate the same gene in different context section and we describe this in the Methods/Variant fine-mapping section. We provide a scatterplot of this for the reviewer below.

6. It is recommended that the abstract include a presentation of the outcome where the overexpression of POP5 or KBTBD6 leads to an increase in telomere length.

The abstract includes a description of this result.

REVIEWERS' COMMENTS

Reviewer #2 (Remarks to the Author):

The authors have addressed all of my comments.

Reviewer #3 (Remarks to the Author):

The authors have addressed my previous comments adequately.

Reviewer #4 (Remarks to the Author):

I am happy to see the authors addressed my major concern of the weak link between the computational results in the sections of statistical analysis and the subsequent validation experiments. The authors clarified the quantitative thresholds and filtering steps that were carried out before experimental validation, providing clear and reproducible selection criteria about how the informative elements from the GWAS-related downstream analysis came together and thus provided putative targets for validation. The readability of the manuscript has also been largely improved for readers outside the field. I recommend the manuscript for publication.